# GBCapsNet: A calibrated capsule network for automated gallbladder disease diagnosis via ultrasound imaging

Madhu Golla[1], Hareesha Katiganere Siddaramappa[2*],
Puvvala Jogeeswara Venkata Naga Sai[1], Sai Karthik Adla[1], Chandrika Naga[1],
Pradeep Nijalingappa[3]

**1** Department of Information Technology, VNR Vignana Jyothi Institute of Engineering and Technology, Hyderabad, Telangana, India, **2** School of Computer Engineering, Manipal Institute of Technology, Manipal Academy of Higher Education, Manipal, Karnataka, India, **3** Department of Computer Science and Engineering (Data Science), Bapuji Institute of Engineering and Technology, Davanagere, Karnataka, India

* hareesh.ks@manipal.edu

## Abstract

Gallbladder diseases present a significant clinical challenge due to their diverse manifestations and the difficulty of accurate interpretation in ultrasound imaging. Manual assessment of gallbladder ultrasound images is time consuming, operator dependent and may delay clinical decision making, motivating the development of automated diagnosis approaches. In this study, we propose a customized capsule network architecture, termed GBCapsNet, for multi-class disease classification using ultrasound images. The model incorporates a modified routing mechanism designed to improve feature representation and class discrimination. The proposed architecture was evaluated across multiple training-test splits, demonstrating high classification performance under image level splits (maximum accuracy of 99.91% and AUC of 1.0). However due to the use of image level splitting rather than patient level separation, these results should be interpreted with caution. Further validation using patient level splits and external datasets is required to establish clinical generalizability. To assess the reliability of predicted probabilities, post training calibration was performed using temperature scaling, resulting in reduced Expected Calibration Error (ECE). These results indicate improved alignment between predicted confidence scores and observed outcomes, although broader validation is required to establish generalizability. To the best of our knowledge, this work represents one of the early investigations into the applications of capsule networks for automated gallbladder disease diagnosis from ultrasound images, Overall, the findings suggest that capsule-based architectures are a promising direction for improving automated interpretation of gallbladder ultrasound data, warranting further validation on larger and more diverse clinical datasets.

**Data availability statement:** The data used in this study are openly available in the UIdataGB dataset at https://data.mendeley.com/datasets/r6h24d2d3y/1 (Turki A, Obaid M, Bellaaj H, Ksantini M, AlTaee A. Gallbladder Diseases Dataset. Mendeley Data, V2. 2024. doi:10.17632/r6h24d2d3y.2).

**Funding:** The author(s) received no specific funding for this work.

**Competing interests:** The authors have declared that no competing interests exist.

## Introduction

The gallbladder is a minute, pear shaped organ located under the liver. The main and initial function of the gallbladder is to store and concentrate bile, a liquid formed by the liver that aids in the digestion of fats. In spite of its relatively simple structure, the gallbladder is prone to multiple disorders such as gallstones, cholecystitis, perforation and polyps. The gallstones are formed when the bile in the gallbladder hardens, forming crystal-hard deposits. These stones can range in size from as small as a grain to as big as a golf ball [1]. These are a major risk factor that can significantly increase the risk of gallbladder cancer, with epidemiological studies indicating that 80% of gallbladder cancer patients present with concurrent gallstones [2]. When gallstones block the bile ducts, it causes irritation, swelling, and inflammation in the gallbladder. This condition is called Acute Cholecystitis [3] and causes severe pain. The complication of acute cholecystitis may lead to Gangrenous Cholecystitis [4], a life-threatening condition due to intense inflammation of the gallbladder that affects the gallbladder tissue, and due to the lack of blood, the tissue begins to die. Another life-threatening complication of acute cholecystitis is perforation of the gallbladder [5], which can happen when the gallbladder wall ruptures due to high inflammation, leading to leakage of the bile. Gallbladder polyps [6] are small growths on the inner lining of the gallbladder that project into the lumen (the space inside the gallbladder). Most polyps are not cancerous, but some have the potential to lead to gallbladder cancer. Early diagnosis of these conditions, including gallbladder wall thickening, adenomyomatosis, carcinoma and associated retroperitoneal complications [7,8] is critical to prevent irreversible damage and improve patient outcomes.

Ultrasound (US) [9], computed tomography (CT) [10], and magnetic resonance cholangiopancreatography (MRCP) [11] are common modalities for diagnosing gallbladder diseases. However, ultrasound is the most widely used imaging modality and a non-invasive procedure commonly recommended by radiologists and physicians for diagnosing gallbladder diseases [9]. Ultrasound imaging is preferred due to its safety, accessibility, and lack of radiation exposure [12]. The accurate diagnosis of gallbladder disease ultrasound images that also presents similar symptoms are carefully analyzed by highly skilled medical specialists to ensure accurate diagnosis and further treatment planning [13]. Though, manual intervention is time consuming and might be vulnerable to human error, several studies have proved that detecting and identifying small sized or non hardened gallstones is quite challenging at times and holds the potential for false positives [14]. Hence, integration of AI into automating the diagnosis of gallbladder conditions is essential for reducing manual intervention, improves accuracy and decreases latency as well as false positive and negative rates to a large extent.

The most recent advancements in neural networks, particularly CNNs, have played a fundamental role in a various computer vision techniques that includes training of image classification models [15], image segmentation models [16] and object detection and identification model use cases [17]. These advancements have motivated and enhanced the potential of researchers to apply CNNs for gallbladder diagnosis using ultrasound images, achieving remarkable performance [18]. As

CNNs have some limitations with respect to the use of Maxpooling makes them inefficient at preserving hierarchical and spatial information in the images and CNNs also require a large dataset for effective training that sometimes might be a limitations for some use cases [19]. To address these limitations, Geoffrey E. Hinton et al. [20] introduced a Capsule Network that groups neurons and uses dynamic routing between capsules along with a routing algorithm for weight updates without backpropagation, making it efficient in preserving hierarchical spatial information. To the best of our knowledge, this study represents the first application of Capsule Networks for gallbladder diagnosis, which is our key contribution.

Although the deep learning models have achieved tremendous accuracy in automating of real time applications. Model calibration refers to the concomitant evaluation between confidence scores and the real likelihood of a model being correct and thus it has become a wide challenge. Poor calibration which is expressed as under confidence or over confidence will lead to decision in errors and serious consequences. This research main objective is to develop a robust automated AI based gallbladder diagnosis system by addressing three important gaps: (1) existing limitations with traditional manual diagnostic methods, (2) inherent constraints of conventional CNNs and (3) the need for model calibration which enhances the model performance in terms of reliability and confidence in inference stages. Not only improved diagnosis but also reliable estimation of confidence is achieved through systematic calibration techniques used in our research. The salient features of the proposed works are as follows:

- To automate the process of gallbladder diagnosis with integration of AI using ultrasound images as input by reducing manual intervention, time consumption, false positives and negatives.

- To mitigate the limitation of CNN and leverage the strengths of the capsule network in preserving hierarchical spatial information for the gallbladder disease diagnosis use case.

- To enhance the true likelihood of the trained model by integrating model calibration techniques into our base model that improves the interpretability of the overall model's prediction confidence.

In the field of deep learning recent studies reveal the advancements in this field have significantly transformed medical imaging analysis, enabling automated detection, segmentation and classification across a range of imaging modalities. This study presents the application of deep learning that prompts key aspects in the diagnosis of the gallbladder. Pang S, et al. [21] proposed a YOLOv3 model for the detection of cholelithiasis (gallstones), the authors used the CT scan modality to classify gallstone types. They introduced the first cholelithiasis CT scan dataset which comprises of 223,846 CT scan images collected from 1369 patients. The proposed YOLOv3 architecture model attained an accuracy of 92.7% and 80.3% in classifying granular and muddy gallstones.

New methodology for segmentation of gallbladder and gallstones in ultrasound images was proposed by Lian, J., et al. [22]. Their model achieved DICE scores of 86.01% and 79.81% for gallbladder and gallstone segmentation respectively. T. Song et al. [23] proposed a segmentation model, U-NeXt, that is used for segmenting gallstones in CT images and also introduced a new dataset for gallstone segmentation consisting of 5350 images collected from 726 patients. The proposed model achieved an intersection over union (IoU) score of 86.97%.

Z. Zhang et al. [24] focused on analyzing clinical features that help differentiate gallbladder cancer diagnosed pre-operatively and unsuspected gallbladder cancer (UGC). The authors first employed a chi-square test to analyze the differences, and then used a random forest algorithm for classification, achieving an accuracy of 73.1%. A study by Urman, J.M., et al. [25] aimed to characterize bile composition in patients with biliopancreatic diseases and identify biomarkers to assist in differentiating biliary strictures. The authors employed data augmentation to generate synthetic data, selected key biomarkers (proteins or metabolites), and analyzed them using neural networks. Their panels of lipids and proteins performed with higher accuracy in differentiating normal patients from cancer patients.

Tao Chen et al. [26] proposed a computer-aided diagnosis system for the segmentation of the gallbladder polyp region, followed by PCA and the Adaboost algorithm to classify the segmented polyp as neoplastic or non-neoplastic. The authors

used high-resolution ultrasound images, and the model achieved an accuracy of 87.54%. Zhou, Qiao-Mei, et al. [27] investigates the radiological characteristics and clinical data of patients with Xanthogranulomatous cholecystitis (XGC) and gallbladder cancer (GBC) to develop a diagnostic model that can distinguish between XGC and GBC. The authors developed three diagnostic models with three different types of imaging data: CT imaging, MRI imaging, and combined CT/MRI imaging. Key features were selected using random forest, and the diagnostic model was developed using a logistic regression model to differentiate XGC from GBC. The CT diagnostic model achieved 83.7% accuracy, the MRI diagnostic model achieved 84.2% accuracy, and the CT/MRI diagnostic model achieved 90.6% accuracy, respectively.

S. I. Jang et al. [28] presented a deep learning-based AI algorithm for differentiating polypoid lesions of the gallbladder (distinguishing between neoplastic and non-neoplastic) using Endoscopic ultrasound (EUS) images. The authors used the ResNet50 pretrained model for the classification task, achieving an accuracy of 89.8%.

A study by Chih-Jui Yu et al. [29] focused on improving gallstone detection and localization of acute cholecystitis using ML. The authors collected over 89,000 abdominal ultrasound images from 2,389 patients and developed two deep learning models. The first model employs ResNet50 as the feature extractor and Single-Shot Multibox Detector (SSD) combined with a Feature Pyramid Network (FPN) as the classifier for gallstones, achieving an AUC of 0.92 and an average inference time of 21 ms. The second model is based on MobileNet V2, a lightweight pretrained model, used for classifying cholecystitis, which achieved an AUC of 0.94 and had an average inference time of 7 ms.

The study by Chang, Y., et al. [30] explores the effectiveness of combining tumor markers with deep learning for diagnosis and prognosis of gallbladder carcinoma. The authors utilized a back-propagation neural network optimized with a genetic algorithm to enhance the detection accuracy of serum tumor markers (CA242, CA199, CA125, and CEA), achieving an accuracy of 94.94%.

C. Loukas et al. [31] proposed a multiple instance learning (MIL) approach to classify gallbladder wall vascularity as low or high using images from laparoscopic cholecystectomy (LC) operations. The authors used a dataset consisting of 181 gallbladder images from 53 surgeries. They designed two classification tasks: image-based classification and video-based classification, using variational Bayesian Gaussian mixture models (VBGMM) and SVM. The image classification model attained an accuracy of 92.1%, however, the video classification model achieved an accuracy of 90.3%.

Zhou, W., et al. [12] built an ensemble deep learning model for diagnosing biliary atresia (BA). The authors used ultrasound images of the gallbladder, achieving a patient-level sensitivity of 93.1% and specificity of 93.9%.

Kim, T. et al. [32] introduced an ensemble model consisting of 3 CNN layers with 5-fold cross-validation for separating true polyps in ultrasound images of the gallbladder. They collected ultrasound images from 501 patients confirmed with gallbladder polyps via cholecystectomy from two tertiary hospitals. The model achieved an accuracy of 87.61%, specificity of 88.35%, and AUC of 0.9082.

H. Fujita et al. [33] proposed a deep learning model to differentiate XGC and GBC using CT images of the gallbladder by collecting CT images from 28 GBC patients and 21 XGC patients and trained a model using a residual CNN with 5 fold cross validation. The model achieved an accuracy of 98.5%, sensitivity of 98.8% and a specificity of 98.0%.

A. M. Obaid et al. [34] developed a deep learning framework for diagnosing biliary atresia (BA) using ultrasound images of the gallbladder. They developed four types of models, including MobileNet, ResNet152, Inception V3, and VGG16, with MobileNet outperforming at 97.87% accuracy, 98.18% sensitivity, and 97.51% specificity.

V. A. and S. Gowrishankar [35] developed a healthcare informatics system using deep learning approaches to detect gallstones in ultrasound images of the gallbladder. They compared several cutting-edge object detection models and found that Mask R-CNN with a ResNet-101-FPN backbone outperformed the other algorithms in accurately detecting gallstones.

The study by G. Cotter et al. [36] aimed to enhance the prognosis for gallbladder cancer patients by leveraging ML to stratify patients based on preoperative factors. The authors collected data from a multi-institutional database of patients who underwent curative-intent resection of GBC, employing Classification and Regression Tree (CART) to stratify patients

based on preoperative factors related to total survival. Based on key factors, patients were stratified into four groups, with median overall survival decreasing progressively across the groups, demonstrating the effectiveness of the model in stratifying patients.

Obaid, A.M., et al. [37] presented a deep learning paradigm for classifying gallbladder diseases using ultrasound images. They developed four deep learning models for detecting nine gallbladder disease types, with the MobileNet model outperforming others at 98.35% accuracy.

Alazwari et al. [38] presented an automated detection system for gallbladder cancer detection with transfer learning in ultrasound images. They employed an inception unit for feature extraction, utilizing the AGTO algorithm for hyperparameter tuning. Finally, they adopted BiGRU for classifying gallbladder cancer, achieving train and test accuracy of 96.28% and 96.29%, respectively.

Putra et al. (2025) [39] developed a gallbladder disease classification framework using CNN-based feature extraction coupled with optimized machine learning classifiers, achieving strong diagnostic performance.

Nabil et al. (2025) [40] introduced MSFE-GallNet, a multi scale CNN model with built in explainability mechanisms for accurate and interpretable gallbladder disease analysis from ultrasound images.

Jayanthi et al. (2026) [41] proposed a hybrid squeeze and excitation capsule network combined with CNN-BiLSTM to achieve high accuracy multi class gallbladder disease diagnosis from ultrasound images.

## Methods

### Dataset overview

The dataset comprises nine distinct classes representing various gallbladder diseases. These samples are collected from an open-source dataset [42], there are 10,692 ultrasound images taken from 1782 patients (UIdataGB). This includes many disease types related to the GB consisting of Gallstones: The tiny cholestrol and calcium crystals forming inside the Gall Bladder, Cholecystitis: Inflammation of the gallbladder, usually caused by bile obstruction and infection, Gangrenous cholecystitis: Severe cholecystitis leading to gallbladder tissue death due to ischemia, Perforation of GB: A life-threatening rupture of the gallbladder wall, often from severe inflammation, polyps and cholesterol crystals: Lipid or tissue growths inside the gallbladder, mostly benign but occasionally precancerous, Gallbladder-wall thickening: Thickening of the gallbladder wall from chronic inflammation or irritation, Adenomyomatosis of the GB: Benign gallbladder wall overgrowth with mucosal epithelial hypertrophy and Luschka's crypts, Carcinoma: Malignant tumor originating from the gallbladder lining, potentially spreading to other organs, Intraabdominal and Retroperitoneum problems: Conditions affecting the abdominal or retroperitoneal spaces, detectable via imaging are organized around nine important anatomical landmarks each with their corresponding number of samples and patients. A complete description of the dataset is given in Table 1. Fig 1 visualizes the sample images from the dataset. Initially, all images are resized to 128×128 before feeding them into the network. 90% of the images are used for training the model, and 10% of the images are used for testing the model. We followed a generalization split of 90−10–50−50. These splits were performed at the image level due to data set constraints and hence generalization for real world might be overestimated and should be implemented as preliminary. Table 2 illustrates the distribution of gallbladder data samples used in this study for training and testing.

The customized capsule network architecture is being depicted in this section that is designed as a three block system for enhanced gallbladder disease classification: (1) Convolutional block, (2) Capsule block and (3) Calibration block, as shown in Fig 2. The input data is initially processed through a series of convolutional layers that are used to bring out the inter-class variations and complex visual features from the gallbladder ultrasound images fed as the input to the model. The processing of such extracted features happens through navigation of blocks and layers while preserving their spatial orientations through the routing mechanism between the primary and secondary capsules optimized using a margin loss function. Then, the calibration block evaluates how well the model's confidence matches with real outcomes to improve the model's interpretability. Table 3 describes the parameters used in the proposed architecture.

**Table 1. Detailed description of the dataset.**

| Class No. | Disease Type | No. of Samples | No. of Patients |
|---|---|---|---|
| 0 | Gallstones | 1326 | 221 |
| 1 | Abdomen and retroperitoneum | 1170 | 195 |
| 2 | Cholecystitis | 1146 | 191 |
| 3 | Membranous and gangrenous cholecystitis | 1224 | 204 |
| 4 | Perforation | 1062 | 177 |
| 5 | Polyps and cholesterol crystals | 1020 | 170 |
| 6 | Adenomyomatosis | 1164 | 194 |
| 7 | Carcinoma | 1590 | 265 |
| 8 | Gallbladder wall thickening (various causes) | 990 | 165 |

Table notes: Dataset collected from UIdataGB [42].

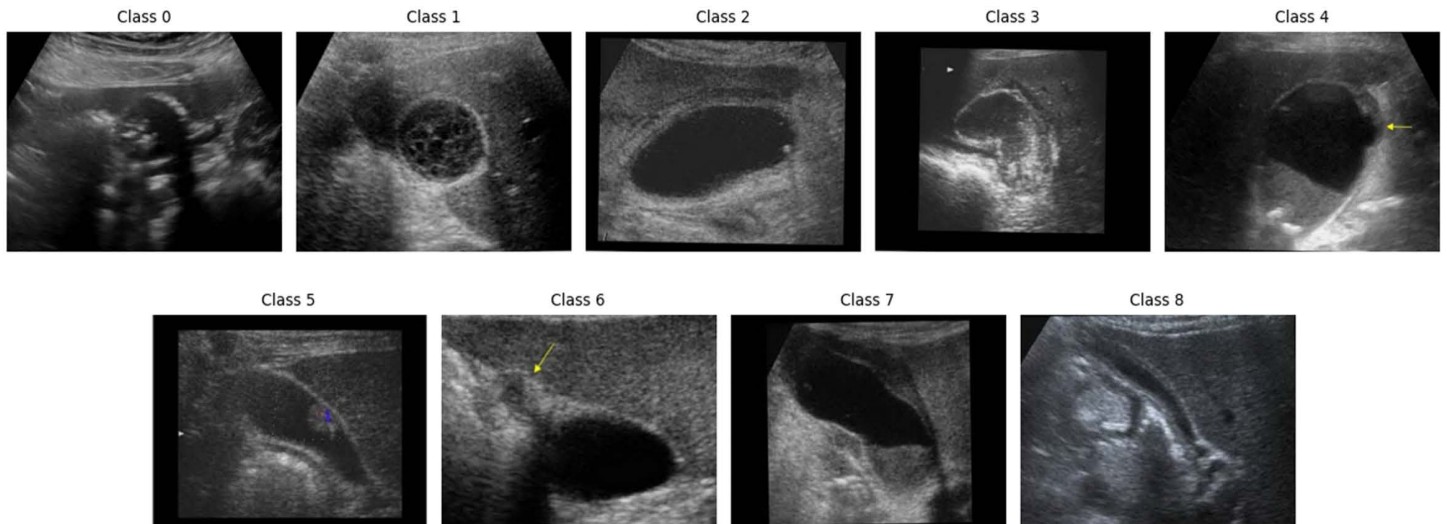

**Fig 1. Illustration of sample images from the dataset [42].**

**Table 2. Distribution of data for generalization.**

| Generalization Split | Training Samples | Testing Samples |
|---|---|---|
| 90–10 | 9622 | 1070 |
| 80–20 | 8553 | 2139 |
| 70–30 | 7484 | 3208 |
| 60–40 | 6415 | 4277 |
| 50–50 | 5346 | 5346 |

Table notes: Training-testing splits used for model generalization testing.

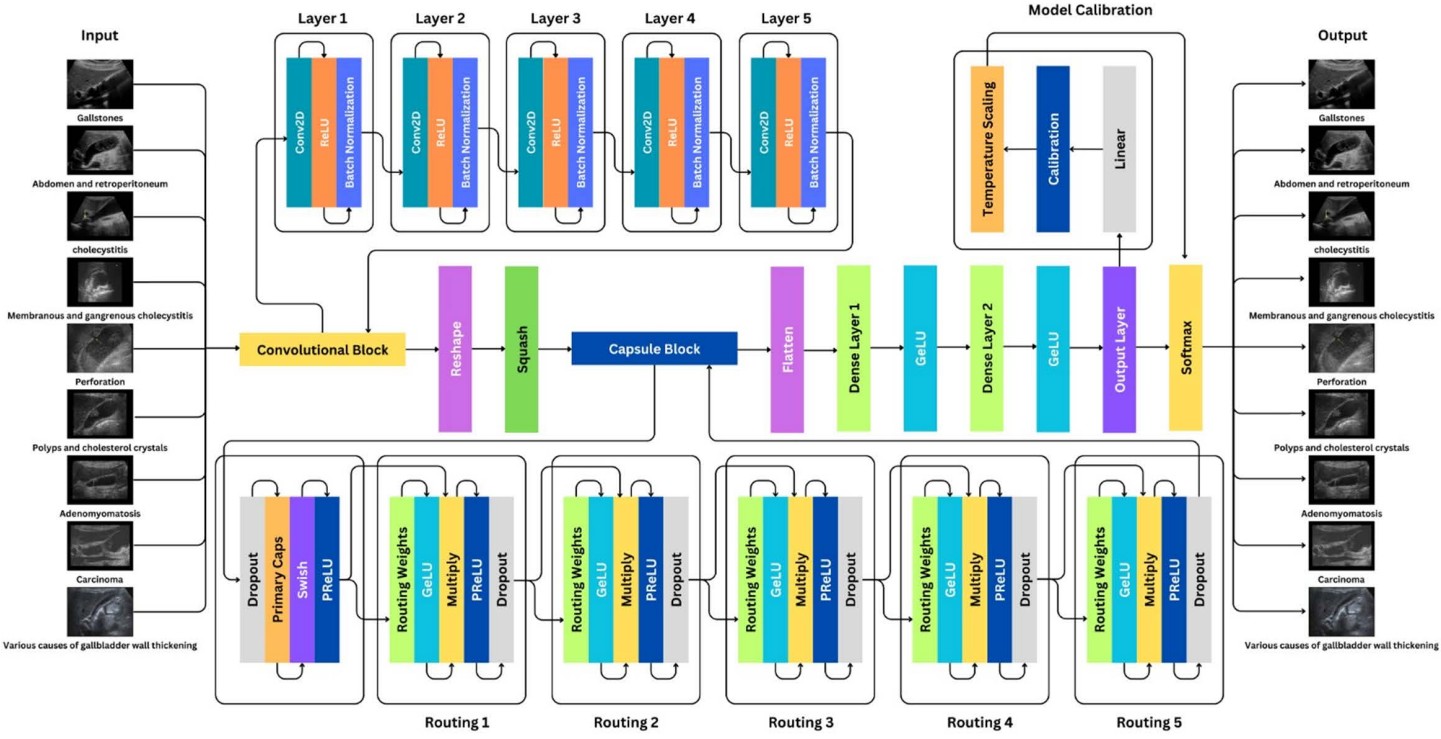

**Fig 2. Architecture of the proposed workflow.**

## Convolutional block

The proposed model employs a convolutional neural network (CNN) for initial feature extraction, as illustrated in Fig 2. In this step, the input data is fed into a series of convolutional layers to extract an array of feature maps. The convolutional block consists of five sequential layers, each followed by a ReLU activation function (Eq. (1)) [43] and batch normalization at each stage.

$$\text{ReLU}(x) = \max(0, x). \tag{1}$$

The max pooling layer aids in achieving translation invariance but loses the spatial hierarchy. It emphasizes whether an object exists rather than its location. The use of max pooling restricts CNN from generalizing well to new poses of objects, as it discards spatial relationships [20].

## Capsule block

In this block, the features extracted by the CNN layer are grouped to form primary capsules, which are lower-level capsules and represent the various instantaneous properties of the objects in the image, such as orientation, position, size, and spatial relations. The extracted features are reshaped into vectors using the reshape function. The next step is a nonlinear squash function (Eq. (2)), is applied to ensure the length of each vector in the output vector is [0,1], which represents the probability of the entity's presence.

**Table 3. Illustration of parameters used in the proposed model.**

| Parameters | Values |
|---|---|
| Batch Size | 32 |
| Dropout | 0.25 |
| Epochs | 100 |
| Kernel Initializer | LeCun normal |
| Learning Rate | 1e-03 |
| Loss Function | Margin Loss |
| Activation Function | ReLU |
| Optimizer | Adam |
| **CNN Layer 1** | |
| Number of Filters | 32 |
| Kernel Size | 3x3 |
| Stride | 2 |
| Padding | same |
| **CNN Layer 2** | |
| Number of Filters | 16 |
| Kernel Size | 3x3 |
| Stride | 2 |
| Padding | same |
| **CNN Layer 3** | |
| Number of Filters | 64 |
| Kernel Size | 3x3 |
| Stride | 2 |
| Padding | same |
| **CNN Layer 4** | |
| Number of Filters | 32 |
| Kernel Size | 3x3 |
| Stride | 2 |
| Padding | same |
| **CNN Layer 5** | |
| Number of Filters | 64 |
| Kernel Size | 3x3 |
| Stride | 2 |
| Padding | same |

Table notes: Hyperparameters and architecture details of GBCapsNet model.

$$S_v = \frac{\|I_v\|^2}{1 + \|I_v\|^2} \times \frac{I_v}{\|I_v\| + \epsilon}.$$

(2)

Here, $\mathbf{S}_v$ is the vector output of the capsule $v$ and $I_v$ is its total input. The input capsule $I_v$ is the weighted sum of the predictor vector $\hat{\mathbf{p}}_{j|i}$, which is computed from the output of the previous capsule layer by multiplying the length of the pose vector $\mathbf{p}_i$, as described in Eq. (3).

$$\hat{p}_{j|i} = W_{ij} * p_i,$$

(3)

$$I_v = \sum_i c_{ij} * \hat{p}_{j|i}.$$

(4)

In Eq. (4), $c_{ij}$ is the coupling coefficient, which is determined by the iterative dynamic mechanism. These primary capsules aim to predict the output of the higher-level capsules, known as secondary capsules. This interaction between primary and secondary capsules is referred to as dynamic routing. During this process, secondary capsules assess the predictions made by primary capsules and compute the agreement among them. Capsules making correct predictions are assigned higher weights, while those making incorrect predictions receive lower weights. This method of weight assignment is known as routing by agreement. The customized routing mechanism used in this research is shown in Table 4 along with the routing process that was followed for successful and efficient model training. The margin loss function (Eq. (5)) is utilized to penalize incorrect predictions. The final output vectors from the capsule layer are then forwarded to a feedforward network for additional processing and training. Each layer in the feedforward network employs GELU as an activation function (Eq. (6)) [44].

$$\text{Margin}_{\text{loss}} = y_{\text{true}} \times \max(0, 0.9 - y_{\text{pred}})^2$$

$$+0.45 \times (1 - y_{\text{true}}) \times \max(0, y_{\text{pred}} - 0.1)^2,$$

(5)

$$\text{GeLU(Input)} = 0.5 \times \text{Input} \times \left(1 + \tanh\left(\sqrt{\frac{2}{\pi}}\left(\text{Input} + 0.044715 \times \text{Input}^3\right)\right)\right).$$

(6)

### Introduction to calibration rationale and references

The study by Chuan Guo et al. [45] highlighted the problem of confidence calibration, which depicts how well a model's predicted confidence aligns with the actual likelihood of correctness. The authors discovered that modern neural networks are poorly calibrated, either being underconfident or overconfident.

**Table 4. Customized dynamic routing mechanism.**

| Input | $\hat{p}_{i|j}$: Input vector from layer $x$ primary capsule $i$ to layer $(x+1)$ secondary capsule $j$; $r$: Number of routings; $x$: Current layer |
|---|---|
| **Output** | $V$: Refined output vector |
| **Algorithm Steps** | |
| 1 | Initialize $V \leftarrow \hat{p}_{i|j}$ for all primary capsules $i$ in layer $x$ and secondary capsules $j$ in layer $x+1$ |
| 2 | **For $k=1$ to $r$:** |
| 3 | For all primary capsules $i$ in layer $x$: $S_{ij} \leftarrow \text{GELU}\left(\sum \mathbf{X} W_{ij} + b_{ij}\right)$ |
| 4 | For all secondary capsules $j$ in layer $x+1$: $\epsilon_{i|j} \leftarrow S_{ij} \odot \mathbf{X}$ |
| 5 | For all secondary capsules $j$ in layer $x+1$: $V \leftarrow \text{PReLU}(\epsilon_{i|j})$ |
| 6 | **Return $V$** |

Table notes: Dynamic routing algorithm pseudocode for GBCapsNet.

## Explanation of Softmax, logits, and temperature scaling

In a traditional neural network, the Softmax activation (Eq. (7)) is applied to the logits produced by the output layer to obtain the probabilities.

$$\text{Probability}(y = i \mid \mathbf{z}) = \frac{e^{z_i}}{\sum_{k=1}^{n} e^{z_k}}. \tag{7}$$

These probabilities can be overconfident or underconfident. To address this problem, we use temperature scaling, a post-training calibration technique introduced by Chuan Guo et al. [45], on the proposed model. It is an extension of Platt scaling [46], with a single parameter, temperature $T$, which helps in calibrating the model probabilities without changing the actual predictions. The temperature scaling equation is described in Eq. (8) and is applied before SoftMax activation. The logits (raw scores) are divided by the temperature $T$ before applying SoftMax activation.

$$\text{Probability}_T(y = i \mid \mathbf{z}) = \frac{e^{\frac{z_i}{T}}}{\sum_{k=1}^{n} e^{\frac{z_k}{T}}}. \tag{8}$$

The optimal temperature value is calculated by the negative log loss function (NLL) (Eq. (9)) and applied unchanged to test data. The objective is to minimize the NLL.

$$\text{NLL} = \arg\min_{T} \left( -\sum_{i=1}^{j} \log \left( \text{Softmax}(z_i, T) \right) \right). \tag{9}$$

Where $\mathbf{z}$ is a vector of logits $[\mathbf{z}_1, \mathbf{z}_2, \mathbf{z}_3, \ldots, \mathbf{z}_n]$, $n$ is the number of classes, and $T$ is a scalar parameter, whose value is greater than 0.

The initial activation function of the proposed model's output layer was SoftMax and has been replaced with a linear activation to obtain the raw logits, as described in the Model Calibration section in Fig 2. To ensure effective calibration using temperature scaling, sufficient testing data is required. Thus, we experimented with three different train-test splits: 70−30, 60−40, and 50−50, ensuring a minimum of 30% of the data is reserved for testing, which is deemed sufficient for calibrating the model. The findings are illustrated in Table 8.

## Model calibration

Post-training temperature scaling was applied to improve the alignment between predicted probabilities and observed outcomes. The temperature parameter was learned on a subset of the training data, held out for validation, for the 70−30, 60−40, and 50−50 train-test splits. Calibration was not performed for the primary 90–10 split, as the limited size of the test set was deemed insufficient for reliable estimation of calibration metrics.

## Ethics statement

This study used an open-source, fully anonymized gallbladder ultrasound dataset. As no identifiable patient information was used, formal ethics approval was not required.

## Results

The model is trained for five iterations on 90% (9622 samples) and tested on the remaining 10% (1070 samples), using the Adam optimizer with a learning rate of $1 \times 10^{-3}$. Table 5 illustrates the model's performance for five iterative routings

**Table 5. Illustration of model performance for 5 iterative routings.**

| Routing Layer | Accuracy (%) | | Loss (%) | | Time Taken (s) | |
|---|---|---|---|---|---|---|
| | Train | Test | Train | Test | Train | Test |
| Layer 1 | 99.59 | 99.72 | 0.003 | 0.003 | 270.83 | 0.33 |
| **Layer 2** | **99.46** | **99.91** | **0.003** | **0.001** | **314.18** | **0.34** |
| Layer 3 | 99.32 | 98.22 | 0.005 | 0.014 | 338.56 | 0.41 |
| Layer 4 | 99.31 | 98.79 | 0.005 | 0.010 | 362.25 | 0.37 |
| Layer 5 | 99.09 | 98.50 | 0.007 | 0.013 | 363.33 | 0.41 |

Table notes: Performance metrics across iterative routing layers for GBCapsNet.

on the 90−10 train-test split. Fig 3 presents the training dynamics through the accuracy and loss curves for all routings. Based on the results, the model trained with two iterative routings with 99.91% testing accuracy and 0.001% testing loss, as near-perfect performance is observed hence, patient level validations are required before any clinical claims are made. Evaluation metrics of the proposed model with two iterative routings are described in Table 6.

According to our final observation the second iteration of the routing configuration achieved outstanding performance. The complete training process required just 314 seconds (approximately 5 minutes), not compromising and maintaining the fast inference speed of 0.34 seconds per test case is very efficient and robust in the field of Applied AI for gallbladder disease diagnosis. These metrics demonstrate the computational efficiency of the model while delivering high diagnostic accuracy. The strong generalizability of the model is quantitatively demonstrated in Table 7 representing comprehensive evaluation metrics across all test conditions. For qualitative analysis, Fig 4 provides the confusion matrix visualization, revealing the model's classification patterns and occasional errors across the nine diagnostic categories.

Expected Calibration Error (ECE) is a key metric for evaluating how well a model is calibrated [47]. ECE quantifies the difference between predicted probabilities and actual labels over multiple bins. It divides the range of predicted probabilities into several bins and calculates the average predicted confidence and accuracy for each bin. Then, it computes the

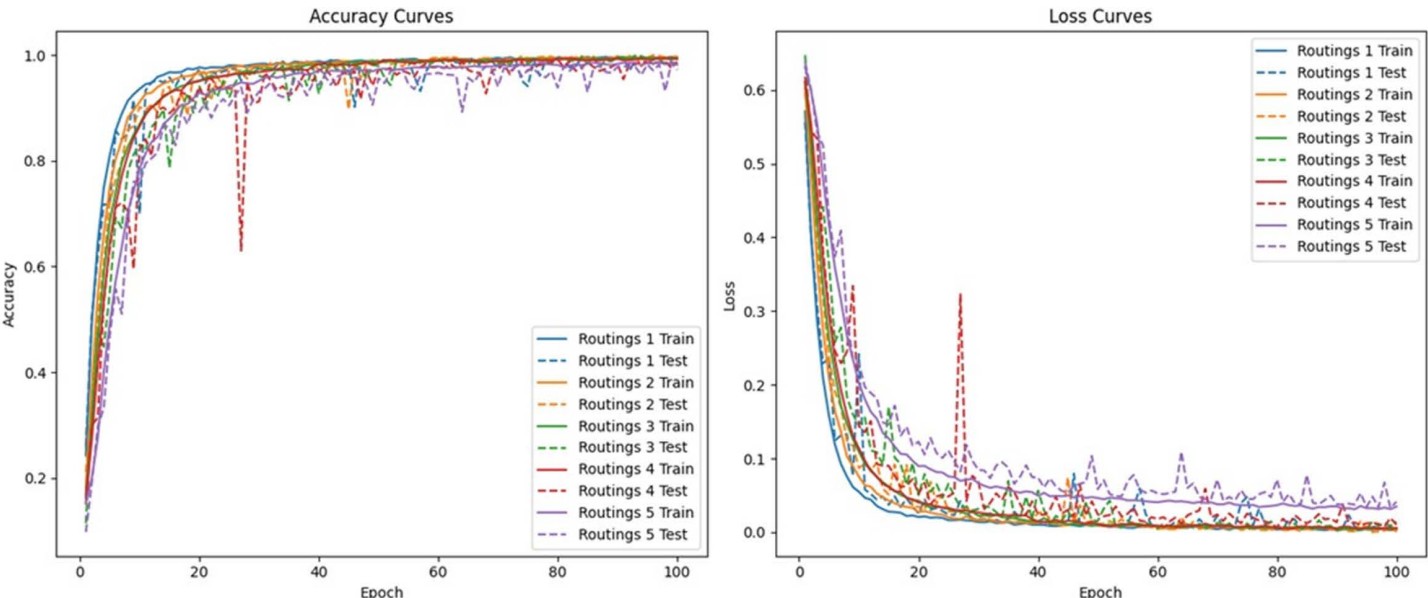

**Fig 3. Architecture learning curve accuracy and loss plots of 5 iterative routings.**

**Table 6. Performance evaluation of two iterative routing models.**

| Class | Precision | Recall | F1-Score | AUC (Train) | AUC (Test) |
|---|---|---|---|---|---|
| 0 | 1.00 | 1.00 | 1.00 | 1.00 | 1.00 |
| 1 | 1.00 | 1.00 | 1.00 | 1.00 | 1.00 |
| 2 | 1.00 | 0.99 | 1.00 | 1.00 | 1.00 |
| 3 | 0.99 | 1.00 | 1.00 | 1.00 | 1.00 |
| 4 | 1.00 | 1.00 | 1.00 | 1.00 | 1.00 |
| 5 | 1.00 | 1.00 | 1.00 | 1.00 | 1.00 |
| 6 | 1.00 | 1.00 | 1.00 | 1.00 | 1.00 |
| 7 | 1.00 | 1.00 | 1.00 | 1.00 | 1.00 |
| 8 | 1.00 | 1.00 | 1.00 | 1.00 | 1.00 |

Table notes: Model evaluation metrics by disease class.

**Table 7. Evaluation of generalization splits on the proposed model with 2 iterative routings.**

| Split | Accuracy (%) | | Loss (%) | | Time Taken (s) | |
|---|---|---|---|---|---|---|
| | Train | Test | Train | Test | Train | Test |
| 90–10 | 99.46 | 99.91 | 0.003 | 0.001 | 314.18 | 0.34 |
| 80–20 | 99.51 | 97.66 | 0.004 | 0.021 | 284.52 | 0.54 |
| 70–30 | 99.33 | 96.82 | 0.006 | 0.027 | 262.32 | 0.72 |
| 60–40 | 98.94 | 93.69 | 0.007 | 0.054 | 255.91 | 1.08 |
| 50–50 | 98.35 | 89.08 | 0.005 | 0.098 | 239.83 | 2.76 |

Table notes: Accuracy, loss, and time over different train-test splits.

weighted average of the absolute difference between accuracy and confidence across all bins. The formula for ECE is described in Eq. (10). The reliability diagrams depicting the relationship between confidence and accuracy across bins for the three train-test splits are illustrated in Fig 5.

$$\text{ECE} = \frac{1}{N} \sum_{i=1}^{m} n(b_i) \times \left| \text{accuracy}(b_i) - \text{confidence}(b_i) \right|. \tag{10}$$

### Findings from calibration

It can be observed from Table 8 that the ECE (%) is increasing when the training samples are decreasing and testing samples are increasing, indicating reduced training data results in poor calibration, also the difference between pre-calibration ECE values and the post-calibration ECE values is increasing, indicating that more testing samples can significantly allow temperature scaling to better calibrate the model. Generally, the temperature value can be > 1 for the models that are overconfident and < 1 for the models that are underconfident. The temperature parameter for calibration was learned on a validation subset derived from the training data and was not tuned on the test set.

From Table 8, all three optimal temperature values are < 1, indicating that the model is underconfident and that temperature scaling has improved the model's confidence. It can be observed that when the training samples and testing samples are equal, the model shows a higher ECE (3.53%) value, indicating poor calibration due to reduced training data. Additionally, at the same time, ECE drops to 0.49% (post-calibration) with a significant difference in ECE between pre-calibration and post-calibration, suggesting that larger testing samples can significantly improve calibration.

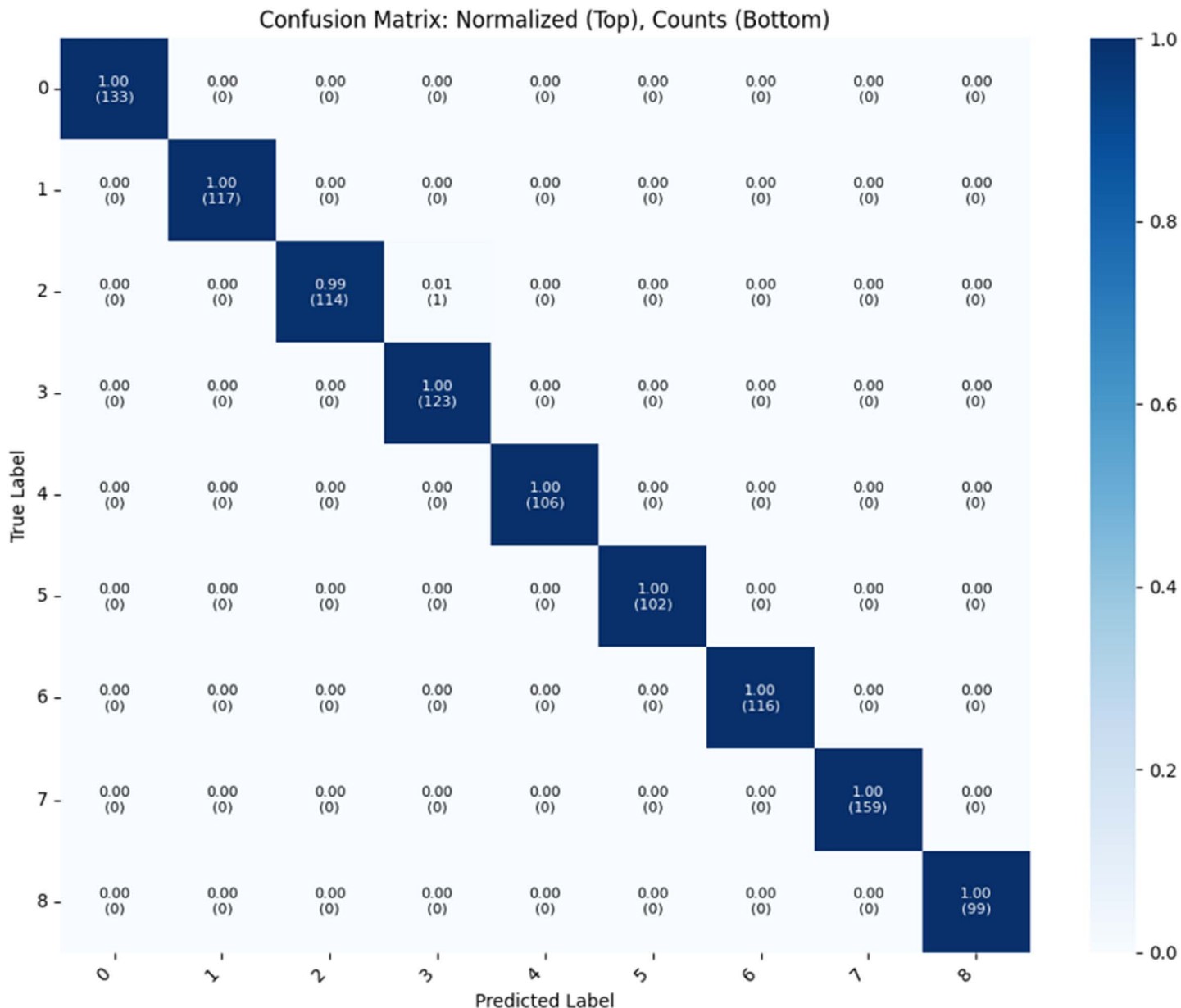

**Fig 4. Confusion matrix of the proposed model with two iterative routings on a 90–10 train-test split.**

## Discussion

The comparative analysis in Table 9 indicates that the proposed capsule network model achieves high classification performance on the evaluated dataset, with an accuracy of 99.91%. However, it is important to emphasize that these comparisons with previously published studies are not conducted under controlled experimental settings, as differences in datasets, data splits, and evaluation protocols can significantly influence reported results. Therefore, such comparisons should be interpreted with caution.

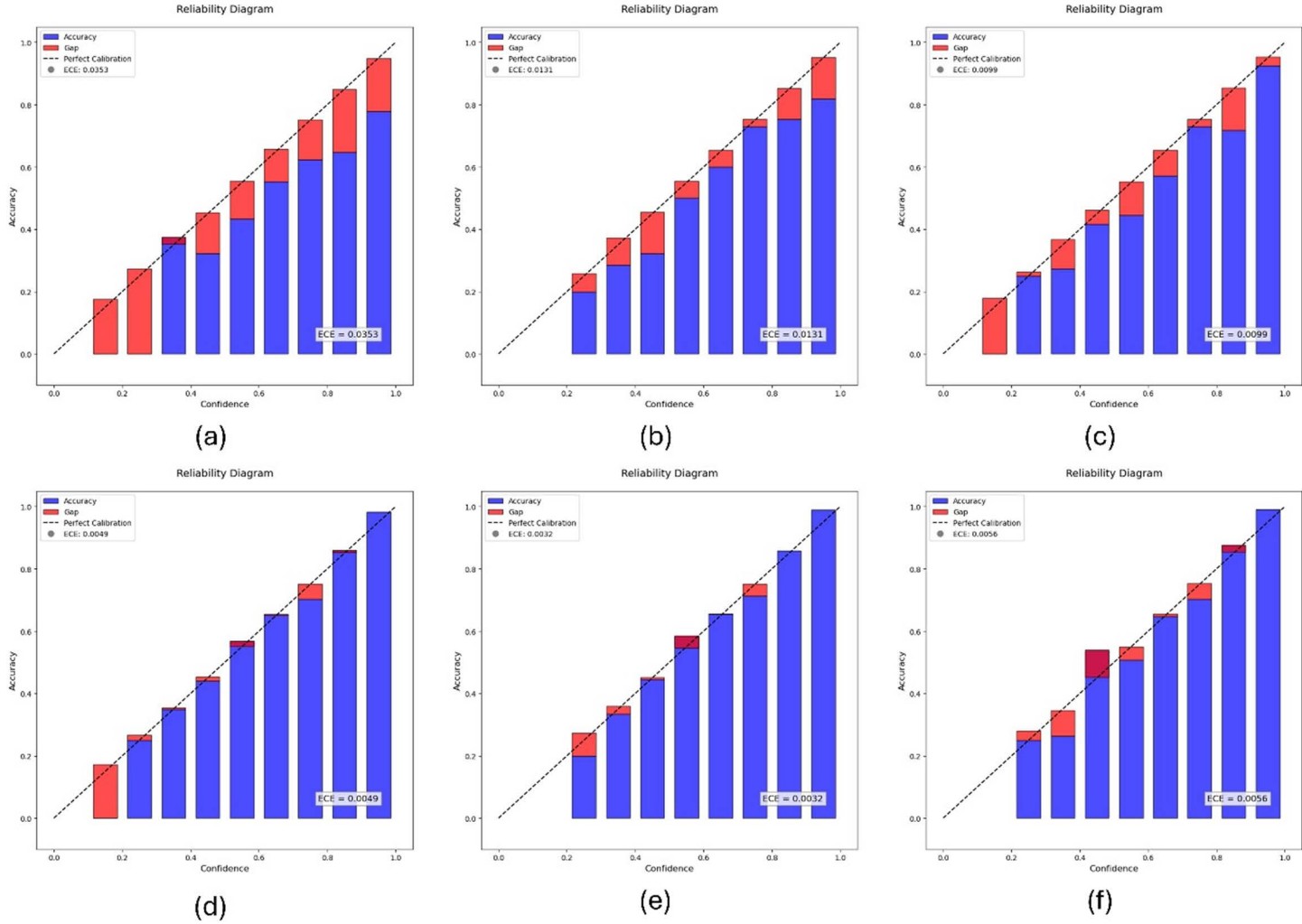

**Fig 5. Reliability diagrams before and after calibration on different train-test splits.** Top row: (a) before calibration (50% testing), (b) before calibration (40% testing), (c) before calibration (30% testing). Bottom row: (d) after calibration (50% testing), (e) after calibration (40% testing), (f) after calibration (30% testing).

Traditional machine learning techniques such as Random Forest, PCA+AdaBoost, and earlier CNN architectures have reported moderate performance (73–95%), while more advanced deep learning models, including ResNet, MobileNet, and ensemble CNNs, have demonstrated higher accuracies (87–98%) under their respective experimental conditions [39–41]. However, direct comparison across studies is not strictly valid without standardized benchmarking on the same dataset and splits.

The high performance observed in this study may be attributed to the capsule network's ability to capture spatial hierarchies and preserve pose information in ultrasound images, combined with optimized routing mechanisms that enhance feature representation across multiple layers. At the same time, it is important to acknowledge that the dataset was split at the image level rather than the patient level, which introduces a potential risk of data leakage due to intra-patient correlations and may result in overly optimistic performance estimates.

**Table 8. Findings of temperature scaling calibration on the test set.**

| Split | ECE (%) Before | ECE (%) After | Difference | Optimal Temperature |
|---|---|---|---|---|
| 70–30 | 0.99 | 0.56 | 0.43 | 0.2675 |
| 60–40 | 1.31 | 0.32 | 0.99 | 0.2992 |
| **50–50** | **3.53** | **0.49** | **3.04** | **0.3473** |

Table notes: Calibration results showing Expected Calibration Error (ECE) before and after temperature scaling, with corresponding optimal temperatures. The reported optimal temperature values were obtained from validation data and applied unchanged to the test set.

**Table 9. Comparative performance analysis of proposed vs. existing models for gallbladder diagnosis.**

| Authors (Year) | Approach | Accuracy (%) |
|---|---|---|
| Z. Zhang et al. (2018) | Chi-square + Random Forest | 73.10 |
| Pang S. et al. (2019) | YOLOv3 neural network | 92.70 |
| Tao Chen et al. (2020) | PCA+AdaBoost | 87.54 |
| Zhou Q.-M. et al. (2021) | Random Forest + Logistic Regression | 90.60 |
| S. I. Jang et al. (2021) | ResNet-50 | 89.80 |
| C. Loukas et al. (2021) | VBGMM+SVM | 92.10 |
| Kim T. et al. (2021) | Ensemble CNN (5-fold CV) | 87.61 |
| Chang Y. et al. (2022) | GA-optimized BPNN | 94.94 |
| H. Fujita et al. (2022) | Residual CNN (5-fold CV) | 98.50 |
| A. M. Obaid et al. (2022) | MobileNet | 97.87 |
| A. M. et al. (2023) | MobileNet | 98.35 |
| Alazwari et al. (2024) | AGTO + Inception + BiGRU | 96.29 |
| Putra et al. (2025) [39] | CNN+ML optimization | 99.35 |
| Nabil et al. (2025) [40] | Optimized CNN Model | 99.63 |
| Jayanthi et al. (2026) [41] | Capsule network + LSTM | 99.09 |
| **Proposed Model** | **Capsule Networks** | **99.91** |

Table notes: Accuracy values from prior studies are reported on different datasets and evaluation protocols; this table is for illustrative purposes only, not for direct comparison.

While the results are promising, they should be interpreted with caution. Further validation using patient-level data partitioning, controlled baseline comparisons (e.g., with standard architectures such as ResNet or MobileNet under identical conditions), and evaluation on independent datasets is necessary to provide a more rigorous assessment of model performance.

Overall, the proposed approach provides a preliminary framework for automated gallbladder diagnosis using capsule-based architectures. However, additional validation is required before drawing conclusions regarding comparative performance or clinical applicability.

## Limitations

The performance of the proposed model depends on image quality, and poor-quality images or rare artifacts may adversely affect diagnostic accuracy. In addition, the dataset was split at the image level rather than the patient level, which introduces a significant risk of data leakage due to correlations between images from the same patient. This may lead to overly optimistic performance estimates and limits the generalizability of the reported results.

Furthermore, no controlled baseline experiments (e.g., using standard architectures such as ResNet or MobileNet under identical conditions) were conducted, which restricts the strength of comparative claims. Nested cross-validation or an explicit validation split would further strengthen the robustness of the calibration analysis.

Future work may include advanced preprocessing techniques and data augmentation strategies to mitigate image quality issues and improve model robustness.

## Conclusion

This research developed an automation framework for gallbladder diagnosis based on a capsule network that utilizes ultrasound images. We compared the performance of five iterative routings of the capsule network, and the model with two iterative routings achieved high classification performance, with a reported testing accuracy of 99.91% under the considered experimental setup. The model achieved a training accuracy of 99.46%, a training loss of 0.003%, and an AUC of 1.0 across all nine classes. Comparisons with previously reported studies are presented in Table 9 however, these are not conducted under controlled experimental settings and should be interpreted with caution. We also tested the alignment between the proposed model's predictions and real values and used temperature scaling for their calibration. The three train-test splits – 70−30, 80−20, and 90−10 – yielded initial ECE values of 0.99%, 1.31%, and 3.53%, respectively. Subsequently, 0.56%, 0.32%, and 0.49% were calibrated using temperature scaling. Temperature scaling improved calibration performance across all evaluated splits. Observations from Table 7 reveal that the optimal temperature of 0.3473 significantly improved confidence calibration between pre- and post-calibration, with a difference of 3.04%. Applying this value before SoftMax activation enables reliable confidence predictions and is incorporated into the web interface, displaying the model's confidence both pre- and post-temperature scaling. However, it is important to note that the dataset was split at the image level rather than the patient level, which introduces a risk of data leakage due to intra-patient correlations and may lead to overly optimistic performance estimates. In addition, no controlled baseline experiments were conducted under identical conditions, limiting the ability to draw definitive conclusions regarding comparative model performance.

Future research should focus on careful integration of the proposed model into clinical workflows, with rigorous validation, as the current study remains in a development phase and has not been clinically validated and also as discussed in the design considerations section, and testing additional gallbladder imaging techniques, including CT scans, to enhance the model's ability to generalize across different imaging modalities. It will also evaluate new routing algorithms between capsules to improve dynamic routing efficiency and representation learning. State-of-the-art pre-trained models can serve as feature extractors to enrich feature extraction. Examining sophisticated model calibration methods can improve the accuracy and understandability of the model in real-time. Extensively augmenting the dataset through various procedures can increase its size and diversity, thus providing additional testing data for calibration methods. Therefore, the findings of this study should be considered preliminary and require further validation on patient-level splits and independent datasets.

## Author contributions

**Conceptualization:** Hareesha Katiganere Siddaramappa.

**Data curation:** Puvvala Jogeeswara Venkata Naga Sai, Sai Karthik Adla.

**Formal analysis:** Madhu Golla, Sai Karthik Adla, Pradeep Nijalingappa.

**Methodology:** Madhu Golla, Hareesha Katiganere Siddaramappa, Chandrika Naga, Pradeep Nijalingappa.

**Software:** Madhu Golla, Puvvala Jogeeswara Venkata Naga Sai, Sai Karthik Adla, Pradeep Nijalingappa.

**Supervision:** Chandrika Naga.

**Validation:** Sai Karthik Adla, Chandrika Naga, Pradeep Nijalingappa.

**Visualization:** Hareesha Katiganere Siddaramappa, Puvvala Jogeeswara Venkata Naga Sai, Sai Karthik Adla, Chandrika Naga.

**Writing – original draft:** Madhu Golla, Hareesha Katiganere Siddaramappa, Puvvala Jogeeswara Venkata Naga Sai.

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
