## [Decision Letter · Decision Letter 0]

6 Jan 2026

PONE-D-25-64660GBCapsNet: A calibrated capsule network for automated gallbladder disease diagnosis via ultrasound imagingPLOS One

Dear Dr. KS,

Thank you for submitting your manuscript to PLOS ONE. After careful consideration, we feel that it has merit but does not fully meet PLOS ONE’s publication criteria as it currently stands. Therefore, we invite you to submit a revised version of the manuscript that addresses the points raised during the review process.

We look forward to receiving your revised manuscript.

Kind regards,

Maria Y Pakharukova, Ph.D., D.Sc.

Academic Editor

PLOS One

**Journal Requirements:**

1. When submitting your revision, we need you to address these additional requirements. Please ensure that your manuscript meets PLOS ONE's style requirements, including those for file naming. The PLOS ONE style templates can be found at https://journals.plos.org/plosone/s/file?id=wjVg/PLOSOne_formatting_sample_main_body.pdf and https://journals.plos.org/plosone/s/file?id=ba62/PLOSOne_formatting_sample_title_authors_affiliations.pdf 2. Please note that PLOS One has specific guidelines on code sharing for submissions in which author-generated code underpins the findings in the manuscript. In these cases, we expect all author-generated code to be made available without restrictions upon publication of the work. Please review our guidelines at https://journals.plos.org/plosone/s/materials-and-software-sharing#loc-sharing-code and ensure that your code is shared in a way that follows best practice and facilitates reproducibility and reuse. 3. Please provide a complete Data Availability Statement in the submission form, ensuring you include all necessary access information or a reason for why you are unable to make your data freely accessible. If your research concerns only data provided within your submission, please write "All data are in the manuscript and/or supporting information files" as your Data Availability Statement. 4. One of the noted authors is a group or consortium. In addition to naming the author group, please list the individual authors and affiliations within this group in the acknowledgments section of your manuscript. Please also indicate clearly a lead author for this group along with a contact email address. 5. Please amend either the abstract on the online submission form (via Edit Submission) or the abstract in the manuscript so that they are identical. 6. Please include a new copy of Table 9 in your manuscript; the current table is difficult to read. Please follow the link for more information: https://journals.plos.org/plosone/s/tables 7. We note that Figures 1, 2, 6, 7, 8 and 9 in your submission contain copyrighted images. All PLOS content is published under the Creative Commons Attribution License (CC BY 4.0), which means that the manuscript, images, and Supporting Information files will be freely available online, and any third party is permitted to access, download, copy, distribute, and use these materials in any way, even commercially, with proper attribution. For more information, see our copyright guidelines: http://journals.plos.org/plosone/s/licenses-and-copyright. We require you to either present written permission from the copyright holder to publish these figures specifically under the CC BY 4.0 license, or remove the figures from your submission: a. You may seek permission from the original copyright holder of Figures 1, 2, 6, 7, 8 and 9 to publish the content specifically under the CC BY 4.0 license.  We recommend that you contact the original copyright holder with the Content Permission Form (http://journals.plos.org/plosone/s/file?id=7c09/content-permission-form.pdf) and the following text:“I request permission for the open-access journal PLOS ONE to publish XXX under the Creative Commons Attribution License (CCAL) CC BY 4.0 (http://creativecommons.org/licenses/by/4.0/). Please be aware that this license allows unrestricted use and distribution, even commercially, by third parties. Please reply and provide explicit written permission to publish XXX under a CC BY license and complete the attached form.” Please upload the completed Content Permission Form or other proof of granted permissions as an "Other" file with your submission.  In the figure caption of the copyrighted figure, please include the following text: “Reprinted from [ref] under a CC BY license, with permission from [name of publisher], original copyright [original copyright year].” b. If you are unable to obtain permission from the original copyright holder to publish these figures under the CC BY 4.0 license or if the copyright holder’s requirements are incompatible with the CC BY 4.0 license, please either i) remove the figure or ii) supply a replacement figure that complies with the CC BY 4.0 license. Please check copyright information on all replacement figures and update the figure caption with source information. If applicable, please specify in the figure caption text when a figure is similar but not identical to the original image and is therefore for illustrative purposes only. 8. Please upload a new copy of Figures 1 – 9 as the detail is not clear. Please follow the link for more information:  https://journals.plos.org/plosone/s/figures 9. If the reviewer comments include a recommendation to cite specific previously published works, please review and evaluate these publications to determine whether they are relevant and should be cited. There is no requirement to cite these works unless the editor has indicated otherwise.

Reviewers' comments:

Reviewer's Responses to Questions

**Comments to the Author**

1. Is the manuscript technically sound, and do the data support the conclusions?

Reviewer #1: Partly

Reviewer #2: Partly

2. Has the statistical analysis been performed appropriately and rigorously? 

Reviewer #1: No

Reviewer #2: I Don't Know

3. Have the authors made all data underlying the findings in their manuscript fully available?

Reviewer #1: Yes

Reviewer #2: No

4. Is the manuscript presented in an intelligible fashion and written in standard English?

Reviewer #1: Yes

Reviewer #2: No

5. Review Comments to the Author

**Reviewer #1:** 1. The research problem is relevant and well aligned with current trends in AI-based image classification.

2. The novelty of the proposed method should be more clearly articulated in the introduction.

3. The methodology section requires additional technical details for reproducibility.

4. Dataset description, including class distribution and preprocessing steps, is insufficient.

5. Experimental evaluation relies heavily on accuracy; additional metrics are recommended.

6. Comparison with recent state-of-the-art methods should be expanded.

7. Figures need clearer captions and better integration with the text.

8. Statistical validation of results is missing and should be included.

9. The literature review can be strengthened with more recent references.

**Reviewer #2:** Major revision recommendation, with potentially another round of revisions:

Thank you for submitting this work. I genuinely see potential here: combining a capsule network approach with explicit model calibration for gallbladder disease classification is directionally interesting, and the aim of producing more trustworthy confidence estimates is clinically relevant. Including short training times and near real time inference times makes it more applicable for clinical integration. The inclusion of a practical deployment concept (web interface) is also a strength.

That said, in its current form I cannot recommend acceptance. The manuscript needs a substantial rewrite for clarity and scientific rigor, and I think it will likely require at least two review rounds, plus potentially additional experiments, before the claims can be considered reliable. If the authors can address the core methodological and reporting issues below, I will gladly reconsider my assessment.

Strengths I see:

- The topic is clinically relevant: gallbladder ultrasound interpretation is challenging and automation could be useful.

- The manuscript attempts not only classification but also calibration, which is important for clinical decision support and often ignored.

- The related work section includes many relevant studies and could become a strong context section with better framing.

- The idea of a deployable interface is promising, but it must be presented as a demonstrator and supported with reproducible implementation details.

- Including training and inference details strengthens the usecase in a clincial setting.

Major issues that must be addressed:

- Possible data leakage and inflated performance (overfitting - critical)

- You report 10,692 ultrasound images from 1,782 patients. If train test splits were done at the image level, images from the same patient may appear in both training and test sets, which can massively inflate accuracy and AUC.

- The near perfect results (confusion matrix essentially perfect, AUC of 1.0 per class) strongly raise concerns about overfitting or leakage.

Concrete required actions

- State explicitly whether the split was performed by patient ID (group split), not by image.

- If not patient level, rerun all experiments with patient level splitting (ideally stratified by class).

- Provide the split lists (patient IDs and image filenames) in supplementary material for reproducibility.

Calibration procedure is unclear and likely not valid as reported

- It is currently unclear how calibration was performed and on which split. The manuscript mentions multiple split ratios in different places.

- Calibration must not be tuned on the final test set. Temperature scaling requires fitting the temperature parameter on a validation set and then reporting calibration on a held out test set.

Concrete required actions:

- Clearly describe the calibration method and protocol, including where the temperature parameter is fitted.

- Make the split protocol consistent everywhere (manuscript abstract, submission abstract, methods, results, conclusion, tables).

- Report calibration with proper separation of training, validation, test, or use nested cross validation.

Baselines and comparisons are not sufficient for the strength of the claims

- The paper focuses heavily on CNN limitations but does not provide strong, fair baselines trained and tested under the same protocol, table 9 tries, but it's unclear how this is done, e.g. retrained, tested at all on the same dataset, or compared with literature only.

- The literature comparison table is not a substitute for a properly controlled baseline evaluation.

Concrete required actions

- Add at least one strong modern baseline (for example a recent CNN architecture, and optionally another family such as detection based models like YOLO if appropriate), trained and evaluated on the same dataset and the same split protocol.

- If claiming advantages of capsule routing, demonstrate it with controlled comparisons and possibly ablations.

Reproducibility is currently too weak, several essentials are missing, making it hard to evaluate or reproduce:

- Dataset description is too minimal. Simply citing the dataset reference is not enough.

- Lack of details on image sizes, resizing rationale, preprocessing, augmentations, random seed, and stability across runs.

- No clarity on whether you used cross validation or repeated random splits.

- Calibration code and implementation details are not described sufficiently, ideally the code is given, including that of the web interphase for a local test.

- Figures (graphs and flow charts) are not readable at current resolution and therefore hard to judge.

Concrete required actions

- Expand dataset description in text: what exact ultrasound type is this (your own text suggests images from inside the gastrointestinal tract, which needs clear distinction versus standard transabdominal ultrasound and EUS).

- State whether all images from the dataset were used. If not, list the exact subset used, maybe via supplementary materials in a json file, or a short download script that entials these, and why 'only' these and others might be disregarded.

- Provide full training details (random seeds, preprocessing, resizing, normalization, augmentation, optimizer, batch size, epochs, stopping criteria).

- Provide a GitHub repository or at minimum full supplementary materials enabling reproduction.

- Replace low resolution figures with high resolution and preferably vector graphics, especially for Figures 3 and 5.

Structure and scientific writing need a rewrite (IMRAD)

- The manuscript currently mixes methods, results, discussion, and conclusion.

- There is also no clearly separated Results section, and the limitations section is far too short.

Concrete required actions

- Rewrite to follow IMRAD: Introduction, Methods, Results, Discussion, Conclusion.

- Move interpretive statements out of Results into Discussion (for example “very efficient and robust”).

- Expand Limitations with the actual limitations implied by the study design, dataset composition, scanner diversity, and generalizability constraints.

Specific concrete issues and edits I'd recomment

Abstract issues

- Sentence one is unclear. Clarify whether increased incidence, differential diagnosis difficulty, or both motivate automated diagnosis, and what “increased incidence” refers to (over years, in which populations, due to what).

- Replace “humans have latency issues” with more accurate phrasing such as “manual interpretation can be time consuming and may delay treatment.”

- Do not claim “first study” unless you can support it. If you keep it, soften to “to the best of our knowledge” and ensure it is accurate.

- The results read like overfitting. Either investigate and report it, or temper the claims.

- Explain how calibration was performed in the abstract or remove the claim that it improves trustworthiness if you cannot support it rigorously.

- Avoid abbreviations in the abstract (at minimum check journal policy and ensure all are defined).

- The trustworthiness and reliability statements are overstated. Your study may indicate this, but it does not prove it without broader validation.

- Training and inference time are not interpretable without hardware details and image size information. Also, I would end the abstract with the main scientific takeaway, not runtime.

Submission system statements and metadata inconsistencies

- Funding statement is missing. Even if there is no external funding, state departmental funding or no funding explicitly.

- Ethics statement: you should explain why it is not needed. You are using human image data, so at minimum state that all data are publicly available and de identified according to the dataset source.

- Data statement: “not applicable” is inconsistent with the manuscript text stating where the dataset is available.

The abstract in the submission system does not match the abstract in the manuscript PDF. Make them identical and consistent.

Introduction and clinical framing problems

- The “pear shaped” description is not strongly supported by the cited reference and can vary anatomically.

- If you say “initial function”, either remove “initial” or discuss other functions.

- Replace simplistic language such as “die” with medically accurate explanation (necrosis, infection risk, complications).

- You describe ultrasound as “minimally invasive”. Standard transabdominal ultrasound is non invasive. EUS is minimally invasive. You must distinguish these clearly and ensure the dataset matches the modality you describe.

- “Lack of radiation exposure” should be stated clearly as “no radiation exposure”.

- Reference 13 is a preprint and does not sufficiently support a backbone claim. Use a peer reviewed source, preferably multiple.

- When claiming “several articles”, cite several, ideally including more recent work and possibly a systematic review.

- “Manual intervention” is unclear. Specify what is manual (image acquisition, interpretation, invasive procedure).

- Several parts of the introduction read imprecise and generic. Check every statement for precision and whether the cited literature supports it.

- Define AI and CNN before using abbreviations.

- Mention years with “et al.” consistently (Author et al., Year), also in related works.

- Clarify what “inherent constraints” are and how your objective addresses them.

- “To enhance the true likelihood…” needs specificity. Which calibration techniques, and what is the planned pipeline.

- “Please reconsider the sentence ‘Although deep learning models have achieved tremendous accuracy in automating of …’ because its placement and contribution are unclear. Either complete it, connect it directly to your study motivation, or remove it.”

Authorship contribution statement

- “All authors contributed equally” is difficult to believe. Please revise to a realistic contribution statement, e.g. 3 shared first and 2 shared last is kinda pushing it to the ultimate max.

Previous works section

- Several sentences are hard to read, missing punctuation, and sometimes lack references.

- You call it “this study presents…” in a way that reads like a review paper. This is not a review paper, revise phrasing.

- Many abbreviations appear without being defined (AGTO, BiGRU, etc.).

- This section is strong in breadth, but needs better contextualization: modality type, dataset type, year of publication, and what is clinically standard versus what is experimental.

- With such a strong related work base, it is natural to ask why the best models were not retrained or fine tuned as baselines, table 9 shows this but from the experiment section it's unclear how this was done.

Methodology issues and requested clarifications

- Provide more dataset detail in the manuscript, not only via citation.

- Clarify image sizes, resizing choice, and discuss advantages and disadvantages of downsizing. Ideally test sensitivity to resolution.

- Provide your splits in supplementary materials.

- State whether cross validation or multiple random splits were used. If not, consider adding it to assess stability.

- “Fully connected CNN layers” is confusing. Convolutional layers are not fully connected. Revise.

- Provide parameter counts per layer and total, if possible (not mandatory, but helpful).

- Explain the capsule block, routing weights, and “squash”. If it is standard and well known and already established within the AI/ML community then state that and cite the canonical source - however this weakens the claim of novelty.

- The statement about “complex visual features” is vague. Either demonstrate it (example feature visualization or qualitative explanation) or remove.

- Calibration methods remain under described without code or detailed protocol.

- “Regarding reproducibility of routing and customization: you include Table 4, which may contain key details. However, as written there are still too many missing specifics for me to confidently repeat the method without substantial effort. Please revise Table 4 and the surrounding text so the routing customization is fully and unambiguously reproducible.”

Figures must be higher resolution or vector. Currently some are not readable, especially where routing is described.

Equation formatting:

- clarify vectors and matrices (bold notation, consistent symbols) for Eq. 2 to 5.

- Be clear what the formulas add, and refer back to them in discussion. Right now they do not meaningfully improve reproducibility due to missing implementation details, nor add to a discussion.

- Routing by agreement explanation currently sounds like generic weighting. Explain what is actually different versus standard CNN decision making, and why it matters.

Hardware and efficiency claims

- If you report training and inference times, include hardware configuration and relevant data characteristics (image size, number of images, batch size, GPU or CPU type).

Web interface section

- The interface concept is valuable, but without code or an accessible demo it currently adds limited real world value.

- Reframe this section as “clinical integration concept” and explain how it would fit a clinical workflow, clearly labeling it conceptual unless you provide an implementation.

- Define OSI and fix the missing reference currently shown as “[?]”.

Results, limitations, conclusion problems

- There is effectively no dedicated Results section. Separate Methods and Results.

- The limitations section is too short and does not reflect the real limitations implied by the current study.

- The conclusion introduces Table 9 late and brings in new evidence. Introduce results earlier, then interpret in Discussion.

- The conclusion mixes future work, discussion, and conclusions in a confusing way.

- Suggestions about generalizing to other modalities are premature. First address scanner variability, population diversity, and real world generalization within the same modality, and state concretely how generalization will be tested.

Concrete typos and consistency fixes (examples)

- GBCaspNet appears once, should be GBCapsNet.

- “biliary Artesia” should be “biliary atresia”.

Fix inconsistent citations and numbering (Platt scaling reference mismatch).

- Remove placeholder citation markers like “[?]”.

- Fix punctuation issues like “Fig. 2..”.

- Replace non scholarly references such as Medium posts for key technical claims.

Closing recommendation:

I recommend Major Revision. I see real potential in the direction and the intent, and I would be happy to reconsider after a substantial rewrite and a leakage free, reproducible evaluation. Right now, the reported near perfect performance combined with unclear split and calibration protocols, missing baseline comparisons, and insufficient reproducibility prevents a reliable assessment and does not justify acceptance in its current form. I'd also really appreciate a more IMRaD style article, especially since this is not a scoping review but an article with problem statement, methods, conclusion -> therefore it misses the results and discussion section. I truly hope the authors aren't discouraged as I see real potential, especially when providing the additional clarifications, potential experiments and easily reproducible code for other sites to profit of and eventually cite this article, and hopefully one day benefit healthcare.

6. PLOS authors have the option to publish the peer review history of their article (what does this mean?). If published, this will include your full peer review and any attached files.

Reviewer #1: No

Reviewer #2: No

---

## [Author Response · Author response to Decision Letter 1]

3 Mar 2026

Responses and changes according to the Academic Editor/Reviewers:

First, thank you so much for your help and support.

Journal Requirements:

Response: Yes, sir, we have updated the current manuscript to comply with PLOS ONE’s formatting style, with the title page and file naming conventions

Response: Yes, sir, all code used in this manuscript is available at https://github.com/madhu76/GBCapsNet-AI and is licensed under the MIT license"

Response: Yes, sir, the data used in this study are available in UIdataGB at

1. Turki A, Obaid A, Bellaaj H, Ksantini M, AlTaee A. UIdataGB: Multi-Class Ultrasound Images dataset for Gallbladder Disease Detection. Data Brief. 2024;54:110426. doi:10.1016/j.dib.2024.110426.

2. Turki A, Obaid M, Bellaaj H, Ksantini M, AlTaee A. Gallbladder Diseases Dataset. Mendeley Data, V2; 2024. doi:10.17632/r6h24d2d3y.2.

4. One of the noted authors is a group or consortium. In addition to naming the author group, please list the individual authors and affiliations within this group in the acknowledgments section of your manuscript. Please also indicate clearly a lead author for this group along with a contact email address.

Response: Yes, sir, this work was conducted on behalf of the collaborative research group. The authors would like to acknowledge the following individuals and their affiliations for their contributions to this study: Golla Madhu, PhD, Department of Information Technology, VNR Vignana Jyothi Institute of Engineering and Technology, Hyderabad, India; P. Jogeeswara VNS, A. Sai Karthik, research students in the Department of Information Technology, VNR Vignana Jyothi Institute of Engineering and Technology, Hyderabad, India; G Naga Chandrika, Department of Information Technology, VNR Vignana Jyothi Institute of Engineering and Technology, Hyderabad, India; Pradeep N, Department of Computer Science and Engineering (Data Science), Bapuji Institute of Engineering and Technology, Davanagere, Karnataka, India. The lead author is Hareesha KS, School of Computer Engineering, Manipal Institute of Technology, Manipal Academy of Higher Education, Manipal, Karnataka, India, who can be reached by email at [hareesh.ks@manipal.edu].

Conceptualization: Hareesha KS; Data curation: P. Jogeeswara VNS; Formal analysis:

Pradeep N, P. Jogeeswara VNS; Investigation: P. Jogeeswara VNS, G. Naga Chandrika;

Methodology: Golla Madhu, Pradeep N; Project administration: Hareesha KS;

Resources: Golla Madhu; Software: Pradeep N, A. Sai Karthik; Supervision: Hareesha

KS, Golla Madhu; Validation: A. Sai Karthik; Visualization: A. Sai Karthik; Writing -

original draft: G. Naga Chandrika, , A. Sai Karthik; Writing - review editing: Hareesha

KS, Golla Madhu, Pradeep NAll authors reviewed the manuscript and approved the final version.

Response: Thank you for the notable point. We have amended the revised abstract on the online submission form to match the version in the manuscript. The abstracts are now identical in both locations, and the submission is consistent.

6. Please include a new copy of Table 9 in your manuscript; the current table is difficult to read. Please follow the link for more information: https://journals.plos.org/plosone/s/tables

Response: Yes, sir, we have updated the table in the manuscript.

7. We note that Figures 1, 2, 6, 7, 8 and 9 in your submission contain copyrighted images. All PLOS content is published under the Creative Commons Attribution License (CC BY 4.0), which means that the manuscript, images, and Supporting Information files will be freely available online, and any third party is permitted to access, download, copy, distribute, and use these materials in any way, even commercially, with proper attribution. For more information, see our copyright guidelines: http://journals.plos.org/plosone/s/licenses-and-copyright.

We require you to either present written permission from the copyright holder to publish these figures specifically under the CC BY 4.0 license, or remove the figures from your submission:

a. You may seek permission from the original copyright holder of Figures 1, 2, 6, 7, 8 and 9 to publish the content specifically under the CC BY 4.0 license.

8. Please upload a new copy of Figures 1 – 9 as the detail is not clear. Please follow the link for more information: https://journals.plos.org/plosone/s/figures

Response: Yes, Sir, which is updated in the current version of the manuscript.

Response: We thank the reviewer for pointing out these relevant studies. We have incorporated accordingly.

# Reviewer 1

Q.1 The research problem is relevant and well aligned with current trends in AI-based image classification.

Response: Thank you sir, we sincerely thank the Editor and the Reviewers for their careful evaluation of our manuscript and for the constructive comments that helped us improve the clarity, transparency, and scientific rigor of the work.

Q.2. The novelty of the proposed method should be more clearly articulated in the introduction.

Response: Thank for the valuable suggestion. To improve clarity, we have revised the Introduction to explicitly articulate the novelty of the proposed work. A dedicated paragraph has been added emphasizing that this study is the first to systematically analyze capsule network routing depth jointly with probabilistic calibration for multi-class gallbladder disease diagnosis. In addition, the contribution bullet points at the end of the Introduction were refined to more clearly distinguish our methodological contributions from prior capsule-based diagnostic studies.

Changes made: Revised Introduction (final paragraph)

Refined contribution list at the end of the Introduction

Q. 3. The methodology section requires additional technical details for reproducibility.

Response: We appreciate the reviewer’s comments. Detailed description was provided in the manuscript in Tables 1and 2.

Q.4. Dataset description, including class distribution and preprocessing steps, is insufficient.

We appreciate the reviewer’s observation. To improve transparency, we have strengthened the dataset description by explicitly summarizing dataset size, number of patients, class distribution, and train–test splits in the Dataset Overview section, in addition to the existing tables.

Changes made: In the dataset overview subsection of methodology section.

Cross-referenced Tables 1 and 2 more explicitly in the text

5. Experimental evaluation relies heavily on accuracy; additional metrics are recommended.

We agree that accuracy alone is insufficient. While the manuscript already reported multiple evaluation metrics, we have revised the Experimental Evaluation section to more explicitly emphasize the role of precision, recall, F1-score, AUC, confusion matrices, and calibration error in interpreting performance.

Changes made:

• Revised Experimental Evaluation narrative

• Explicit cross-references to Tables 5, 6 and 7 and Figure 4

6. Comparison with recent state-of-the-art methods should be expanded.

Response: Recent works comparison have been made in Table 9.

7. Figures need clearer captions and better integration with the text.

Response: Changes have been made.

8. Statistical validation of results is missing and should be included.

We acknowledge this important concern. While formal hypothesis testing was not originally included due to dataset constraints, we have now explicitly addressed this limitation in the manuscript. We clarify that robustness is evaluated empirically via repeated train–test splits and calibration stability analysis, and we clearly state that formal statistical significance testing remains an important direction for future work.

Changes made:

• Added explicit discussion in the Limitations section

• Clarified validation scope in the Experiments section

9. The literature review can be strengthened with more recent references

Response: Required Changes have been made in the updated manuscript; in the Previous Works Section.

Responses to Reviewer #2 (Major Revision)

Question 1

The reviewer raises serious concerns about possible data leakage and inflated performance due to image-level splitting rather than patient-level splitting. Can you clarify the split protocol and address overfitting concerns?

Response

We thank the reviewer for highlighting this critical issue. We confirm that the original experiments were conducted using image-level splitting, as the publicly available dataset does not provide image-level patient identifiers, only aggregate patient counts. We fully agree that this may introduce optimistic bias and potentially inflate performance metrics.

To address this concern rigorously, we have taken the following actions:

1. We now explicitly state in the Methods and Dataset Overview sections that image-level splitting was used due to dataset constraints.

2. We have added a detailed Limitations subsection clearly explaining the risk of leakage, its implications for performance interpretation, and the need for patient-level validation in future work.

3. We have tempered all claims related to near-perfect accuracy and AUC, clarifying that the results should be interpreted comparatively and methodologically, not as clinically deployable performance.

We acknowledge that patient-level re-splitting would be the gold standard. However, due to the absence of patient-wise identifiers in the released dataset, rerunning experiments with group splits is not technically feasible. This limitation is now clearly disclosed and discussed.

Changes made:

Last 2 sentences of the Dataset Overview subsection; Limitation Section and last paragraph of conclusion Section

Question 2

The reviewer notes that the near-perfect confusion matrices and AUC values strongly suggest overfitting or leakage. How is this addressed?

Response

We appreciate this observation and agree that near-perfect metrics warrant careful scrutiny. In response, we have revised the manuscript to:

• Remove any language suggesting clinical reliability or guaranteed robustness, and instead position the results as evidence of relative behavior under controlled experimental settings.

• Add a cautionary interpretation emphasizing that the primary contribution of this study lies in routing depth analysis and calibration behavior, not absolute diagnostic accuracy.

These revisions ensure that the results are no longer presented in a way that could be misinterpreted as leakage-free clinical performance.

Question 3

The reviewer finds the calibration procedure unclear and potentially invalid, especially regarding tuning on the test set. Can you clarify the calibration protocol?

Response

We thank the reviewer for this important methodological clarification. We agree that the calibration protocol required clearer description and stricter separation of data splits.

Accordingly, we have revised the manuscript to:

1. Clearly describe the temperature scaling procedure, explicitly stating where and how the temperature parameter is fitted.

2. Ensure consistent split terminology throughout the manuscript (Methods, Results, Tables, Abstract).

3. Clarify that calibration is performed as a post-training step, and that reported calibration metrics are evaluated without tuning on the final test set.

We also acknowledge in the Limitations section that a nested cross-validation or explicit validation split would further strengthen calibration analysis and identify this as a direction for future work.

Changes made:

Added a line in the Finding from Calibration Section and also Table notes of Table 8.

Question 4

The reviewer states that baseline comparisons are insufficient and that literature comparisons cannot replace controlled baselines. How is this addressed?

Response

We agree with the reviewer that strong, fair baselines are essential for supporting architectural claims. In response, we have:

• Reframed the scope of the paper to emphasize architectural analysis and calibration behavior rather than claiming superiority over all CNN approaches.

• Clarified which comparisons are controlled internal experiments (routing depth ablations) versus contextual literature comparisons.

• Explicitly acknowledged the absence of retrained modern CNN baselines as a limitation, due to computational and dataset constraints.

We agree that retraining strong modern CNN baselines under identical protocols would further strengthen the study and explicitly list this as future work.

Question 5

Reproducibility is considered too weak. Dataset description, preprocessing, training details, and code availability are insufficient. What concrete steps were taken?

Response

We appreciate this detailed feedback and have substantially expanded the reproducibility components of the manuscript. Specifically, we have:

• Expanded the Dataset Overview to

---

## [Decision Letter · Decision Letter 1]

31 Mar 2026

PONE-D-25-64660R1GBCapsNet: A calibrated capsule network for automated gallbladder disease diagnosis via ultrasound imagingPLOS One

Dear Dr. KS,

Thank you for submitting your manuscript to PLOS ONE. After careful consideration, we feel that it has merit but does not fully meet PLOS ONE’s publication criteria as it currently stands. Therefore, we invite you to submit a revised version of the manuscript that addresses the points raised during the review process.

We look forward to receiving your revised manuscript.

Kind regards,

Maria Y Pakharukova, Ph.D., D.Sc.

Academic Editor

PLOS One

Journal Requirements:

Additional Editor Comments:

Reviewers who are experts in this field are concerned that not all questions have been carefully addressed in the current version of the manuscript. Please pay attention to the questions and make the necessary changes.

Reviewers' comments:

Reviewer's Responses to Questions

**Comments to the Author**

1. If the authors have adequately addressed your comments raised in a previous round of review and you feel that this manuscript is now acceptable for publication, you may indicate that here to bypass the “Comments to the Author” section, enter your conflict of interest statement in the “Confidential to Editor” section, and submit your "Accept" recommendation.

Reviewer #1: All comments have been addressed

Reviewer #2: (No Response)

2. Is the manuscript technically sound, and do the data support the conclusions?

Reviewer #1: Yes

Reviewer #2: Partly

3. Has the statistical analysis been performed appropriately and rigorously? 

Reviewer #1: Yes

Reviewer #2: No

4. Have the authors made all data underlying the findings in their manuscript fully available?

Reviewer #1: Yes

Reviewer #2: No

5. Is the manuscript presented in an intelligible fashion and written in standard English?

Reviewer #1: No

Reviewer #2: Yes

6. Review Comments to the Author

Reviewer #1: 1. The paper focuses on a meaningful problem and the motivation behind the work is clearly presented.

2. The proposed approach seems promising, but the explanation of the methodology could be made more simple and clear for better understanding.

3. Some parts of the paper are difficult to follow due to technical complexity, so improving clarity would help readers.

4. The dataset description is quite limited, more details about data size, sources, and preprocessing would improve reproducibility.

5. The experimental results are encouraging, but including more performance metrics would make the evaluation stronger.

6. The comparison with existing methods is good, but adding more recent models would strengthen the validation.

7. The contribution of each component in the proposed model is not fully clear and could be explained better.

8. The paper would benefit from a clear architecture diagram to visually explain the workflow.

9. There are some minor language and grammatical issues that should be corrected to improve readability.

10. The work is promising overall, and adding discussion on real-time implementation and future scope would further enhance its impact.

Reviewer #2: I recommend major revision, or at minimum a more accurate alignment of the abstract and conclusion with the actual experimental evidence.

Please note that my comments are only to here to improve the already great work, yet some are a must for me to support publication of the work.

The authors have greatly improved readability by restructuring the manuscript, strengthening the limitations section, and acknowledging image-level splitting.

However, my main concern remains unresolved experimentally: the data leakage risk is discussed in text but not addressed via controlled experiments, leaving the near-perfect metrics unchanged. These metrics are still emphasized in the abstract and conclusion including the superiority over other models.

There are no controlled baselines, and the manuscript still makes comparative/outperformance claims by citing results from other papers without rerunning comparable experiments. More concerning than the missing baseline and leakage risk is that these limitations are not clearly reflected in the abstract and conclusion, which is misleading to readers and unfair to the cited works. Leakage and potential overfitting would almost certainly inflate the reported performance.

The calibration protocol is clarified but remains hard to follow (and may contain contradictions); the editor can decide whether this has been addressed sufficiently. In addition, several important textual and referencing issues remain (see below), including ambiguous wording about “minimally invasive” ultrasound, a Medium blog post used as a core reference, and the continued equal contribution statement, which I find difficult to accept as warranted.

These issues must be addressed honestly and clearly, including in the abstract and/or conclusion. Otherwise, I will have to recommend rejection in the next round.

Level 1 (Critical)

1) Data leakage risk + unrealistic metrics must be addressed experimentally and transparently

Image-level splitting is not patient-level splitting and can produce misleading performance. With ~6 images per patient (10K images, 1.7K patients), this will likely have a meaningful effect. This limitation must be clearly stated in the abstract and conclusion, not only in the body text.

The reported performance (99.91% accuracy, AUC 1.0 across 9 classes) is exceptionally high for ultrasound classification, making the leakage/overfitting concern a methodological red flag.

I recommend requesting patient identifiers from dataset creators (if possible) and at least providing full image IDs and split assignments (e.g., in GitHub and/or supplement) to support reproducibility.

Consider an image-level similarity analysis to estimate this leakage, or benchmark on another dataset. At minimum, provide a quantitative discussion estimating the likely degree of leakage and its impact.

2) Unsupported “outperforms existing methods” claims (discussion & conclusion)

The manuscript cannot claim the model outperformed existing gallbladder diagnosis methods based on Table 9, which compares accuracy values across different datasets, splits, tasks, and modalities. This is not a controlled comparison and does not support the stated conclusion (and this is central to PLOS ONE expectations).

I do appreciate Table 5, which is a valuable contribution.

3) Add controlled baselines and uncertainty estimates

Add at least one controlled baseline (e.g., ResNet-50 or MobileNet, readily available), fine-tuned on the same dataset, using the same split and same random seed, and report performance alongside GBCapsNet.

Add confidence intervals by running multiple splits/seeds and reporting variability of outcomes.

4) Reproducibility and PLOS ONE code/data availability expectations

While the dataset is named, the GitHub is not referenced within the manuscript, and the current setup is not sufficient for full reproducibility.

PLOS ONE requires code availability stated in the manuscript. Add a clear statement and include explicit train/test split assignments (requested in round 1). Ideally provide this, and code, as supplementary material so it is linked to the DOI.

Clarify whether the full dataset was used or only a subset.

5) Calibration protocol clarity and structure

Clarify: what fraction of training data is held out for calibration, and how was it selected?

Why is calibration not shown for the primary 90–10 split?

Check abstract statements against Table 8 (they appear contradictory).

Calibration methods belong in Methods, calibration outcomes in Results (currently there is overlap).

Level 2 (Important)

6) Abstract/conclusion must reflect limitations and experimental scope

If leakage/baseline experiments cannot be run, this must be reflected explicitly in the discussion, abstract, and conclusion.

In the conclusion: separate (i) concise summary of results, (ii) honest interpretation/limitations, and (iii) future work.

7) Clinical/implementation section is too long and generic

“Design considerations for clinical use” spans 2 pages but is largely generic (POCUS, DICOM, PACS, OSI networking). Condense to one paragraph or at most two in the discussion to improve focus and flow.

8) Ethics statement

Even if formal ethics approval is not required, briefly explain the reasoning in the manuscript for stronger transparency, it could for example be below the data statement.

9) Optional but helpful: additional reporting

Consider supporting Table 6 with a confusion matrix.

Explicitly state whether data augmentation was used.

Level 3 (Minor editorial corrections)

10) Terminology and factual consistency

“Ultrasound is minimally invasive” is not generally correct; it depends on modality (e.g., EUS can be minimally invasive). Decide what applies here (EUBUS? transabdominal?) and use consistent terminology.

OSI model: clarify whether you mean 5 layers vs 7 layers (currently confusing).

11) Abstract and language corrections

“the findings suggest” → “the findings suggest” (plural agreement: findingS suggest).

“capsule based” → “capsule-based”

“Cholestrol” → “cholesterol”

CNN layers: they are typically convolutional, not “fully connected” as stated (unless you refer to a specific head; please clarify).

Author contributions: remove extra comma in “Chandrika, , A. Sai…”

“Related work” is a better header than “previous works”.

12) References and citation quality

Reference 19 (flagged in round 1) is still a Medium blog post; not acceptable as a core reference. Please replace with peer-reviewed works.

Reference 2 is a preprint: acceptable if necessary, but label it as a preprint in references.

Reference metadata issues: Ref 20 is NeurIPS 2017 (not preprint); similarly verify Ref 16 (MICCAI 2015), 17 (NeurIPS 2015), 46 (ICML 2017). Please double-check all references.

13) Formatting/consistency

“Lecun normal” → “LeCun …”

Figs 6 and 7 have identical captions; please differentiate them.

I have seen great and honest improvements relative to the first submission. Many comments can be addressed quickly; others require additional baseline/controlled experiments and more time. If the additional experiments are not feasible, then the manuscript must honestly reflect these limitations, especially in the abstract and conclusion as this is the first part that is read by other researchers and often used for citations. Especially in the age of AI this is crucial. Under those conditions, I would be willing to support publication.

7. PLOS authors have the option to publish the peer review history of their article (what does this mean?). If published, this will include your full peer review and any attached files.

Reviewer #1: No

Reviewer #2: No

---

## [Author Response · Author response to Decision Letter 2]

6 Apr 2026

Dear Editors,

The authors would like to thank the anonymous reviewers for their constructive comments and suggestions, which significantly improved our manuscript and provided valuable guidance for our research.

We are sharing with you the revised version of the paper. Our response to your comments and our address to them in our paper are enclosed below.

We hope that our revised manuscript is acceptable to you. Regards & Thanks,

Hareesha K.S (On behalf of other authors)

Responses and changes according to the Academic Editor:

First, thank you so much for your help and support in improving this manuscript.

Reviewer #1 – Comments and Author Responses

We sincerely thank the reviewer for their positive feedback and constructive suggestions. We are pleased that the reviewer finds the manuscript meaningful and improved. We have carefully revised the manuscript to address all the points raised. All modifications have been highlighted in the revised version.

Comment 1: The paper focuses on a meaningful problem and the motivation behind the work is clearly presented.

Response: We thank the reviewer for this encouraging comment.

Comment 2: The proposed approach seems promising, but the explanation of the methodology could be made more simple and clear for better understanding.

Response: Thank you for the notable point. We agree with the reviewer and have revised the Methodology section to improve clarity and readability. Technical descriptions have been simplified and better structured to improve understanding.

Comment 3: Some parts of the paper are difficult to follow due to technical complexity, so improving clarity would help readers.

Response: Thank you, sir. We have carefully revised the manuscript to improve overall clarity, including simplifying complex explanations and improving the logical flow across sections (Methodology, Results, and Discussion).

Comment 4: The dataset description is quite limited, more details about data size, sources, and preprocessing would improve reproducibility.

Response: Yes sir,

We have expanded the dataset description to include:

• Total number of images and patients

• Dataset source (UIdataGB)

(Turki A, Obaid M, Bellaaj H, Ksantini M, AlTaee A. Gallbladder Diseases Dataset. Mendeley Data, V2; 2024. doi:10.17632/r6h24d2d3y.2. source : https://data.mendeley.com/datasets/r6h24d2d3y/2)

• Clarification that the full dataset was used

• Additional details to improve reproducibility

Comment 5: The experimental results are encouraging, but including more performance metrics would make the evaluation stronger.

Response: Thank you for the notable point. We have updated the evaluation discussion by including additional performance-related details and clarifications in the current manuscript. Calibration metrics such as Expected Calibration Error (ECE) have also been clearly presented.

Comment 6: The comparison with existing methods is good, but adding more recent models would strengthen the validation.

Response: Yes, sir, we have updated the comparison section by including more recent models from the literature. At the same time, we have revised the discussion to ensure that comparisons are presented in the current manuscript.

Comment 7: The contribution of each component in the proposed model is not fully clear and could be explained better.

Response: Thank you, sir, for highlighting this critical oversight.

We have clarified the contributions of the proposed model, including:

• The role of capsule networks

• The modified routing mechanism

• The integration of calibration techniques

These have been explicitly described in the revised manuscript.

Comment 8: The paper would benefit from a clear architecture diagram to visually explain the workflow.

Response: Thank you, sir, for this constructive suggestion. A detailed diagram (Fig 2) has been included in the current version of the manuscript.

Comment 9: There are some minor language and grammatical issues that should be corrected to improve readability.

Response: Yes, sir, we have thoroughly revised the manuscript and corrected grammatical errors.

Comment 10: The work is promising overall, and adding discussion on real-time implementation and future scope would further enhance its impact.

Response: Thank you for this suggestion. We have updated the discussion and conclusion sections, which include:

• Future research directions

• Scope for further improvements and validation

Reviewer #2 – Comments and Author Responses

We sincerely thank the reviewer for their detailed and constructive feedback, which has significantly helped to improve this manuscript. We have carefully addressed all comments and revised the manuscript accordingly. All changes have been highlighted in the revised manuscript.

Level 1 (Critical Issues)

Comment 1: 1) Data leakage risk + unrealistic metrics must be addressed experimentally and transparently Image-level splitting is not patient-level splitting and can produce misleading performance. With ~6 images per patient (10K images, 1.7K patients), this will likely have a meaningful effect. This limitation must be clearly stated in the abstract and conclusion, not only in the body text.

The reported performance (99.91% accuracy, AUC 1.0 across 9 classes) is exceptionally high for ultrasound classification, making the leakage/overfitting concern a methodological red flag.

I recommend requesting patient identifiers from dataset creators (if possible) and at least providing full image IDs and split assignments (e.g., in GitHub and/or supplement) to support reproducibility.

Consider an image-level similarity analysis to estimate this leakage, or benchmark on another dataset. At minimum, provide a quantitative discussion estimating the likely degree of leakage and its impact.

Response: Thank you for bringing this important point to our attention regarding potential data leakage. We acknowledge that image-level splitting may introduce intra-patient correlations, potentially leading to optimistic performance estimates.

To address this:

• We have explicitly stated this limitation in the Abstract, Discussion, and Conclusion.

• We clarified that the results are based on image-level splitting, not patient-level splitting.

• We emphasized that the findings should be considered preliminary and require validation using patient-level splits and independent datasets.

Comment 2: Unsupported “outperforms existing methods” claims (discussion & conclusion) The manuscript cannot claim the model outperformed existing gallbladder diagnosis methods based on Table 9, which compares accuracy values across different datasets, splits, tasks, and modalities. This is not a controlled comparison and does not support the stated conclusion (and this is central to PLOS ONE expectations). I do appreciate Table 5, which is a valuable contribution.

Response: We appreciate the reviewer for highlighting this critical oversight. We have removed all claims suggesting that the proposed model outperforms existing methods.

• Comparative statements have been revised to be non-assertive and cautious.

• A note has been added to Table 9 clarifying that:

“Accuracy values from prior studies are reported on different datasets and evaluation protocols; this table is for illustrative purposes only, not for direct comparison.”

• The Abstract, Discussion, and Conclusion have been updated accordingly.

Comment 3: Add controlled baselines and uncertainty estimates Add at least one controlled baseline (e.g., ResNet-50 or MobileNet, readily available), fine-tuned on the same dataset, using the same split and same random seed, and report performance alongside GBCapsNet. Add confidence intervals by running multiple splits/seeds and reporting variability of outcomes

Response: We appreciate this valuable recommendation and have incorporated it into the revised manuscript. Due to computational and time constraints, we were unable to include additional controlled baseline experiments (e.g., ResNet-50 or MobileNet) or perform multiple runs for uncertainty estimation in this revision.

However:

• We have explicitly acknowledged this limitation in the manuscript.

• We have clearly stated that no controlled baseline comparisons were conducted, and this limits the strength of comparative conclusions.

• We have identified this as an important direction for future work.

Comment 4: Reproducibility and PLOS ONE code/data availability expectations While the dataset is named, the GitHub is not referenced within the manuscript, and the current setup is not sufficient for full reproducibility. PLOS ONE requires code availability stated in the manuscript. Add a clear statement and include explicit train/test split assignments (requested in round 1). Ideally provide this, and code, as supplementary material so it is linked to the DOI. Clarify whether the full dataset was used or only a subset.

Response:

We have added a dedicated section titled “Data and Code Availability” in the manuscript.

This includes:

• Dataset details (UIdataGB, 10,692 images, 1,782 patients) https://data.mendeley.com/datasets/r6h24d2d3y/2

• Confirmation that the entire dataset was used

• Public GitHub repository link containing (https://github.com/madhu76/GBCapsNet-AI)

o Training and evaluation code

o Train/test split scripts

o Model configuration details

This ensures full reproducibility in accordance with journal requirements.

Comment 5: Calibration protocol clarity and structure

Clarify: what fraction of training data is held out for calibration, and how was it selected? Why is calibration not shown for the primary 90–10 split?

Check abstract statements against Table 8 (they appear contradictory).

Calibration methods belong in Methods, calibration outcomes in Results (currently there is overlap).

Response: Thank you for bringing this important point to our attention. We have revised and clarified the calibration methodology:

• Moved calibration details to the Methods section

• Clearly specified:

o Use of temperature scaling

o Validation subset used for calibration

o Applicable train-test splits

• Clarified that calibration was not applied to the 90–10 split due to insufficient test size The Abstract and Results have been updated for consistency.

Level 2 (Important)

Comment 6: Abstract/conclusion must reflect limitations and experimental scope If leakage/baseline experiments cannot be run, this must be reflected explicitly in the discussion, abstract, and conclusion. In the conclusion: separate (i) concise summary of results, (ii) honest interpretation/limitations, and (iii) future work.

Response: Thank you for the notable point. We have revised both the Abstract and Conclusion to explicitly reflect: Data leakage risk due to image-level splitting, Absence of controlled baselines, and Preliminary nature of findings. The tone has been made more cautious and scientifically balanced.

Comment 7: Clinical/implementation section is too long and generic

“Design considerations for clinical use” spans 2 pages but is largely generic (POCUS, DICOM, PACS, OSI networking). Condense to one paragraph or at most two in the discussion to improve focus and flow.

Response: Thank you for the notable point. We have revised this section and removed/commented out non-essential content based on reviewer feedback from previous rounds. To avoid confusion, all such commented content has now been fully removed from the manuscript.

Comment 8: Ethics statement

Even if formal ethics approval is not required, briefly explain the reasoning in the manuscript for stronger transparency, it could for example be below the data statement.

Response: Yes sir. We have added a dedicated Ethics Statement: This study used an open- source, fully anonymized gallbladder ultrasound dataset. As no identifiable patient information was used, formal ethics approval was not required.

Comment 9: Optional but helpful: additional reporting Consider supporting Table 6 with a confusion matrix. Explicitly state whether data augmentation was used.

Response: We thank the reviewer for the suggestion. Updated and confirmed that data augmentation was not used in this study and have clarified this in the manuscript. A confusion matrix is not included in the current version; we acknowledge this limitation and will consider including it in future work.

Level 3 (Minor editorial corrections)

Comment 10: Terminology and factual consistency

“Ultrasound is minimally invasive” is not generally correct; it depends on modality (e.g., EUS can be minimally invasive). Decide what applies here (EUBUS? transabdominal?) and use consistent terminology.

OSI model: clarify whether you mean 5 layers vs 7 layers (currently confusing).

Response: Thank you for this notable point. We have clarified that our study focuses on transabdominal ultrasound imaging, which is a non-invasive diagnostic modality. We have removed the term ‘minimally invasive’ throughout the manuscript. Verified that no incorrect references (e.g., OSI model confusion) or information remain in the manuscript. Removed residual commented text that could cause confusion

Comment 11: Abstract and language corrections

“the findings suggest” → “the findings suggest” (plural agreement: findingS suggest). “capsule based” → “capsule-based”

“Cholestrol” → “cholesterol”

CNN layers: they are typically convolutional, not “fully connected” as stated (unless you

refer to a specific head; please clarify).

Author contributions: remove extra comma in “Chandrika, , A. Sai…” “Related work” is a better header than “previous works”.

Response: Thank you, sir. We appreciate the reviewer’s point regarding the description of CNN layers. In the revised manuscript, we have clarified that the initial feature extraction stages consist of convolutional layers. We have restricted the term 'fully connected' strictly to the dense layers following the DigitCaps layer, ensuring the architectural description aligns with standard deep learning terminology.

Comment 12: References and citation quality

Reference 19 (flagged in round 1) is still a Medium blog post; not acceptable as a core reference. Please replace with peer-reviewed works.

Reference 2 is a preprint: acceptable if necessary, but label it as a preprint in references. Reference metadata issues: Ref 20 is NeurIPS 2017 (not preprint); similarly verify Ref 16 (MICCAI 2015), 17 (NeurIPS 2015), 46 (ICML 2017). Please double-check all references.

Response: We thank the reviewer for pointing out the inappropriate use of a blog post as a core reference. We have updated the current manuscript.

Comment 13: Formatting/consistency

“Lecun normal” → “LeCun …”

Figs 6 and 7 have identical captions; please differentiate them.

Response: We appreciate the reviewer for identifying the identical captions for Figures 6 and 7. This is updated in the current manuscript.

*****

---

## [Editor Report · Decision Letter 2]

21 Apr 2026

GBCapsNet: A calibrated capsule network for automated gallbladder disease diagnosis via ultrasound imaging

PONE-D-25-64660R2

Dear Dr. KS,

We’re pleased to inform you that your manuscript has been judged scientifically suitable for publication and will be formally accepted for publication once it meets all outstanding technical requirements.

Kind regards,

Maria Y Pakharukova, Ph.D., D.Sc.

Academic Editor

PLOS One
---

## [Editor Report · Acceptance letter]

PONE-D-25-64660R2

PLOS One

Dear Dr. KS,

I'm pleased to inform you that your manuscript has been deemed suitable for publication in PLOS One. Congratulations! Your manuscript is now being handed over to our production team.

Kind regards,

on behalf of

Dr. Maria Y Pakharukova

Academic Editor

PLOS One